# EVALAGENT: Discovering Implicit Evaluation Criteria from the Web

**Manya Wadhwa♠, Zayne Sprague♠, Chaitanya Malaviya◇, Philippe Laban♡**
**Junyi Jessy Li♠, Greg Durrett♠**
♠The University of Texas at Austin ◇University of Pennsylvania ♡Microsoft Research
manya.wadhwa@utexas.edu

## Abstract

Evaluation of language model outputs on structured writing tasks is typically conducted with a number of desirable criteria presented to human evaluators or large language models (LLMs). For instance, on a prompt like *"Help me draft an academic talk on coffee intake vs research productivity"*, a model response may be evaluated for criteria like accuracy and coherence. However, high-quality responses should do more than just satisfy basic task requirements. An effective response to this query should include quintessential features of an academic talk, such as a compelling opening, clear research questions, and a takeaway. To help identify these implicit criteria, we introduce EVALAGENT, a novel framework designed to automatically uncover nuanced and task-specific criteria. EVALAGENT first mines expert-authored online guidance. It then uses this evidence to propose diverse, long-tail evaluation criteria that are grounded in reliable external sources. Our experiments demonstrate that the grounded criteria produced by EVALAGENT are often *implicit* (not directly stated in the user's prompt), yet *specific* (high degree of lexical precision). Further, EVALAGENT criteria are often not satisfied by initial responses but they are *actionable*, such that responses can be refined to satisfy them. Finally, we show that combining LLM-generated and EVALAGENT criteria uncovers more human-valued criteria than using LLMs alone.[1]

## 1 Introduction

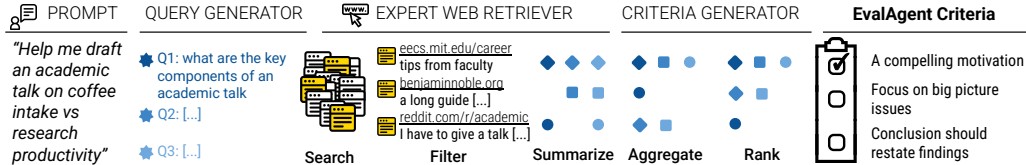

Figure 1: We propose a novel framework EVALAGENT that dynamically generates grounded, implicit evaluation criteria for a given prompt based on retrieved expert advice.

Large Language Models (LLMs) are increasingly used for complex writing tasks, including blog posts, novels, research statements, and more. However, evaluating their outputs remains a significant challenge (Tian et al., 2024; Spangher et al., 2024). The lack of robust, automated evaluation methods limits our ability to scale these evaluations, and hampers our ability to systematically understand and mitigate the errors these models make. Human evaluation has revealed issues such as lack of creativity, concept repetition, and logical inconsistencies (Chakrabarty et al., 2024b). However, scaling human evaluation is subjective, requires expertise, and quickly becomes infeasible given the vast number of tasks LLMs are expected to handle (Chakrabarty et al., 2024a; Kim et al., 2024a).

---

[1]Code and data available at: https://github.com/ManyaWadhwa/EvalAgent

There are two key challenges central to open-ended response evaluation: (1) clearly defining the criteria on which model responses are assessed, and (2) ensuring that evaluators (human or model-based) can reliably interpret and judge these criteria. In this work, we address the first challenge, namely identifying (often implicit) criteria. As the scope of LLM applications expands, it is essential to use evaluation criteria that reflect nuanced writing standards, domain-specific knowledge, and appropriate guidelines.

We propose EVALAGENT, a novel evaluation framework that can scalably generate task-specific evaluation criteria via grounding to expert knowledge from the web. Our proposed framework agentically searches the web to generate implicit task-specific evaluation criteria based on retrieved advice. Figure 1 illustrates our proposed framework. EVALAGENT consists of three key components: (a) a **Query Generator**, which formulates search queries to retrieve instructional web documents, (b) an **Expert Retriever**, which filters the results for good URLs, then extracts and summarizes information from the instruction web documents, (c) a **Criteria Generator**, which synthesizes retrieved information into well-grounded evaluation criteria.

This process mimics the kind of resources that novice humans turn to when attempting to improve content they are producing. In the example from Figure 1, if someone wants a draft for an academic talk, they might search *'how to draft an academic talk'* on a search engine, then navigate to instructional documents like university websites, blogs from academics on how to structure talks, and social media platforms like Reddit that crowd-source advice.

To assess and compare evaluation criteria generated by different systems, we propose three properties that ideal criteria should exhibit: *specificity*, *implicitness*, and *actionability*. *Specificity* ensures that evaluation criteria target precise dimensions of quality, thus reducing ambiguity during assessment. *Implicitness* measures whether we capture the unspoken norms, principles, and conventions within specific writing domains. Finally, *actionability* ensures that criteria facilitate tangible improvements. We evaluate criteria generated from EVALAGENT for these properties across nine datasets consisting a variety of writing tasks, ranging from procedural and technical writing to creative writing. We show that criteria generated by our system are highly actionable and more human-aligned compared to simply prompting an LLM.

Our main contributions are (1) the EVALAGENT framework, demonstrating that we can retrieve implicit evaluation criteria from the web; (2) evaluation of criteria produced by EVALAGENT across several settings, showing how they differ from LLM-generated criteria in that they are more implicit, less obvious, and more actionable.

## 2 Task Setting

Given a user prompt $\mathbf{x}$, LLMs generate a response $\mathbf{y}$ by sampling from a distribution $p(\mathbf{y} \mid \mathbf{x})$. In this work, we propose a dynamic approach for generating a set of evaluation criteria $\mathcal{C}$ tailored to each $\mathbf{x}$.

**Implicit Criteria**   We define an *implicit* criterion as one that a user did not state in their prompt, but which is still specific to the prompt and is a desired property to satisfy. This underspecification can be unintentional, where the user might not be an expert in the task and is not aware of the details to specify. It can also be intentional, where the user assumes certain details are common knowledge or it might feel redundant to state these or they want to save time (Malaviya et al., 2025b). For example, for the user prompt in Figure 1, *'the response should focus on big picture questions'* is an implicit criterion, not specified in the prompt, but important for drafting a good academic talk.

In contrast, *explicit* criteria are often able to be directly derived from the given instruction. Following from the example in Figure 1, explicit criteria for that instruction may be *'the response should be an academic talk'*. We additionally define explicit criteria as unstated but extremely obvious criteria such as *'the response should be focused on the topic'*. Other unstated but extremely obvious criteria may be assuming the response is factually correct or having mathematical rigor for tasks in factuality and mathematics, respectively. Previous work has

explored explicit criteria (Qin et al., 2024); however, our work focuses on generating implicit criteria that is specific, actionable, and closely aligned with human-authored criteria.

**Criteria-Based Scoring**  Once generated, these criteria are used in scoring functions $f(\mathbf{x}, \mathbf{y}, c)$, where each criterion $c \in \mathcal{C}$ to help assess the quality of a response. $f$ can return real-valued scores (Kocmi & Federmann, 2023; Wadhwa et al., 2024a; Kim et al., 2024b; Wu et al., 2025), binary judgments (Olausson et al., 2024; Wadhwa et al., 2024b), or other formats depending on the task (Zhang et al., 2024).

These judgments from $f$ based on $\mathcal{C}$ can be used in multiple ways. For example, they can be aggregated into metrics like pass-rate (e.g., $PR = \sum_{c \in \mathcal{C}} f(\mathbf{x}, \mathbf{y}, c) / |\mathcal{C}|$), or used for response refinement (Cook et al., 2024), or be transformed into dense rewards for training models more effectively (Wu et al., 2023).

Our focus in this work is to identify a set of criteria $\mathcal{C}$ that serve as a shared starting point across multiple applications, such as refinement and evaluation. We envision EVALAGENT to be a flexible framework that enables targeted development of specific capabilities, and we leave the exploration of downstream applications to future work.

## 3  EVALAGENT

In this section, we propose our framework EVALAGENT, aimed at extracting evaluation criteria from instructional web documents. There are several key challenges that make retrieving instructions from the web non-trivial. First, using the prompt as a search query often fails to retrieve URLs that sufficiently cover the criteria for generating a good response. For example, directly searching for prompt in Figure 1 can lead to URLs focused on content of the academic talk, rather than advice on how to structure the talk, so some useful criteria may be missed. Second, generic or low-quality pages might be surfaced; these should be filtered to leave those that are genuinely useful content from authoritative sources. Third, even when relevant information is retrieved, synthesizing the retrieved

**Algorithm 1** Aspect generation with EVALAGENT

---
**Input:** Prompt $\mathbf{x}$
**Output:** Criteria set $\mathcal{C}$
1: $\mathcal{Q} \leftarrow QG(\mathbf{x})$     ▷ Generate set of queries
2: $\mathcal{C}_q \leftarrow \varnothing$
3: **for** $\mathbf{q} \in \mathcal{Q}$ **do**
4:     $\mathcal{U} \leftarrow \text{search}(\mathbf{q}_i)$     ▷ Expert Retriever
5:     $\mathcal{U}' = \text{filter}(U, \mathbf{q}_i, \mathbf{x})$
6:     $\mathcal{C}_{q,i} \leftarrow \varnothing$
7:     **for** $u_k \in U'$ **do**
8:         $r_k = \text{answer}(content(u_k), \mathbf{q}_i, \mathbf{x})$
9:         $\mathcal{C}_{q,i} \leftarrow \mathcal{C}_{q,i} \cup \{r_k\}$
10:     **end for**
11:     $\mathcal{C}_q \leftarrow \mathcal{C}_q \cup \text{summarize}(\mathcal{C}_{q,i})$
12: **end for**
13: $\mathcal{C}_w = AG(\mathcal{C}_q)$     ▷ Criteria Generator
14: $\mathcal{C} = Rank(\mathcal{C}_w, \mathbf{x})$     ▷ Rank
15: **return** $\mathcal{C}$

---

knowledge to clearly identify evaluation criteria relevant to the user prompt is challenging.

EVALAGENT, described in Algorithm 1 comprises several key components designed specifically to overcome the challenges mentioned above. At a high level, given a user promopt, we first generate search queries that can be easily answered using instructional web documents. After retrieving such documents, we generate answers for the the search queries. We then combine these answers to define our evaluation criteria.

**Query Generator (**$QG$**)**  Given the instruction $\mathbf{x}$, we prompt an LLM to generate *conceptual queries* $\mathcal{Q}$ such that each $\mathbf{q}_i \in \mathcal{Q}$ helps retrieve relevant instructional advice for $\mathbf{x}$ (Prompt B.1). These queries help us find targeted advice on concepts useful for the prompt, unlike naively searching for the prompt itself. For example, in Figure 1, the main concept is '*Academic Talk*', a query formulated is: '*what are the key components of an academic talk*'. We also generate queries related to the main concept, even when they are not directly stated in the instruction, for example, '*how to write an engaging talk*'. A similar query expansion step is frequently used in LLM fact-checking (Chen et al., 2024; Malaviya et al., 2024).

**Expert Retriever (**search, filter, summarize**)**  For each $\mathbf{q}_i$, we retrieve a list of URLs ($\mathcal{U}_i$) and filter them based on two properties: (1) expertise of the URL content and (2) the relevance of the URL content to the instruction. We do this filtering to remove shallow, clickbait links and focus on useful advice. We create detailed criteria to evaluate these properties (Appendix

**How to write an engaging fight scene?**

https://lithub.com/how-to-write-a-good-fight-scene/
*A fight scene might use **short sentences or long, quick or slow**. At the climax, **the speed of the scene may in fact slow down, with the character's sense of time expanding.***

answer ▶ ■ Vary the pacing of the fight scene
■ Use short sentences

https://www.dabblewriter.com/articles/how-to-write-a-fight-scene
*The characters you write are just as important to your story as the plot, **so make sure the conflict here is meaningful and has consequences***

answer ▶ ● Weave the characters' backstory
● Establish motivations for conflict
● Write consequences of conflict

summarize ▶ ● Establish characters' backstory
■ Vary the pacing and rhythm to reflect characters emotional states
■ Describe motivations and consequences[…]
**Criteria extracted for this query**

Figure 2: Step 2 of our proposed pipeline, Expert Retriever. Given a query "*How to write an engaging fight scene?*", it retrieves trusted URLs, answers the search query based on the URL content, and summarizes all the query answers to give a query-specific criteria list.

B.1) and prompt an LLM to score the URL content. We sort the URLs by the prompted score and take the top-5 to get $\mathcal{U}'$. For the example in Figure 1, we are able to retrieve and filter to trusted sources of advice from university links and faculty blogs as well as Reddit advice.

For each $u_k \in \mathcal{U}'$, we prompt an LLM to answer $\mathbf{q}_i$ based on $u_k$ (Prompt B.2) and extract $r_k$. $r_k$ is a natural language response that answers $\mathbf{q}_i$. Once we have these answers for all $u_k \in \mathcal{U}'$, we summarize them into a query-specific list, $\mathcal{C}_q$ (Prompt B.3). This is the first step where we re-write the URL content into a list of criteria. This step ensures that we are summarizing only the information relevant to the query and the instruction and have a way of identifying useful information. We illustrate this process for a single query in Figure 2.

**Criteria Generator (***CG***)** Once we have $\mathcal{C}_q$ for all $\mathbf{Q}$, we aggregate them to produce $\mathcal{C}_w$ (Prompt B.4). We illustrate this process in Figure 3. Given query-specific summaries, we aggregate them by putting together common properties seen across summaries and also keeping the unique ones. After aggregation, the criteria go through an instruction-aligned rewriting step such that each point represents a criterion relevant to evaluating $\mathbf{x}$ (Prompt B.5). The rewriting for alignment step is needed to ensure that the generic, instructional web-document advice is made specific for evaluation.

**Ranking** In the final step, a ranking function scores all criteria in $\mathcal{C}_w$ and returns an ordered set of criteria, $\mathcal{C}$. For this work, we score a criterion based on its relevance to $\mathbf{x}$. We use an LLM for this scoring (Prompt B.7). The ordering helps bring the more relevant criteria to the top of the list, which can be useful for downstream applications (e.g. evaluating only using $k$ criteria instead of the entire list).

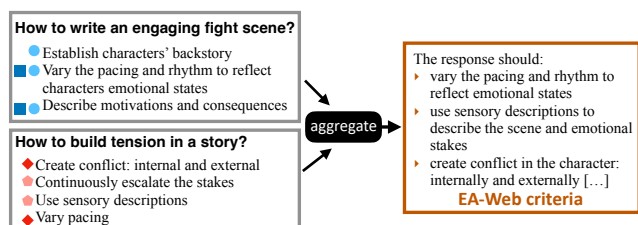

Figure 3: Step 3 of our proposed pipeline, Criteria Generation. We take the criteria generated for each of the queries and then aggregate them using an LLM to distill an evaluation criteria for the instruction.

**Combining with other criteria** EVALAGENT retrieves a set $\mathcal{C}$ of criteria from the web. However, these criteria may not be exhaustive for evaluating the task. In particular, criteria arising directly from the instruction itself are not likely to be captured. As an extension to our system, we can also merge $\mathcal{C}$ with other criteria (e.g., explicit criteria derived from the instruction) to produce a ranked list.

When we only use the criteria derived from the web, we refer to our system as EVALAGENT-WEBONLY (EA-Web). Our full system combines with the LLM-*n* baseline defined below, reranked according to the criteria relevant to the prompt. We refer to this as EVALAGENT-FULL (EA-Full).

We use `gpt-4o-mini-2024-07-18` as the LLM for the components described above. We detail the design choices made for EVALAGENT and present the prompts in Appendix B. We also discuss the computational costs associated with different steps of our proposed framework in Appendix C.

# 4 Metrics for Assessing Generated Criteria

Having established our method, we introduce a set of metrics designed to assess the quality of $\mathcal{C}$. These metrics are novel contributions of this work, as prior research has not systematically evaluated or compared different sets of criteria in a similar manner.

## 4.1 Automatic Metrics

**Specificity ($S$)**    A criterion is specific when it uses less frequently occurring words. We measure specificity of a criterion, $c_i$, as the rarity of words in the criterion over all the sets of criteria generated by the methods we evaluate, $\mathcal{C}^+ = \mathcal{C}_{\text{EA}} \cup \mathcal{C}_{\text{ID}} \cup \mathcal{C}_{\text{LLM-n}}$. Rarity is calculated by the Normalized Inverse Word Frequency (NIWF) (Zhang et al., 2018; Ko et al., 2019). The specificity, $S(\mathcal{C}^+, c_i)$, of a criterion is then defined as the maximum NIWF over the words in the criterion. A higher score indicates that the criterion uses less commonly occurring terms, reflecting a greater degree of specificity.

**Implicitness ($I$)**    A criterion is considered implicit if it surfaces properties not explicitly stated in the instruction. To approximate this, we measure the explicitness or surface-level nature of a criterion through its word-overlap ratio ($WO$) and report 1-$WO$. A higher word-overlap suggests that the criterion closely mirrors the wording of the instruction and is less implicit.

Both $S$ and $I$ are computed for each generated criterion and subsequently aggregated across all criteria produced by a system for a given dataset, providing a holistic view of the system's ability to generate implicit and specific feedback. We report $S$ and $I$ across all instances in the dataset (described in Section 5.1).

**Actionability ($A$)**    A criterion is actionable if the model can effectively revise and enhance its initial response to satisfy the criterion. We test for the actionability of a criterion, $c_i$ by testing if different models can be prompted to satisfy the criteria by revising a response that failed to satisfy $c_i$ initially (Wadhwa et al., 2024b; Madaan et al., 2023). We report $A$ on a sample of 200 instructions. We prompt `gpt-4o-mini-2024-07-18` for evaluating criterion satisfaction (Prompt F.1). Details about the sampling process, dataset distribution and prompts for evaluation are in Appendix F.2.

**Recall of human criteria ($R$)**    We evaluate how frequently human-generated or verified criteria for a given instruction are represented in the generated criteria. To calculate this, we prompt a large language model (LLM) with a human-provided criterion, asking it to determine whether that criterion is entailed by any generated criteria by the system. It is important to note that human written criteria may not fully capture all implicit criteria needed to evaluate a response to an instruction. This lack of coverage means that we cannot treat the human curated list as "gold standard" however, we use it as a proxy for what humans believe is important. We calculate recall by prompting `gpt-4o-mini-2024-07-18`. Prompts for calculating recall and the prompt tuning process are discussed in Appendix F.3.

## 4.2 Human Evaluation

**Utility ($U$):**    A criterion has high utility if it is highly relevant for evaluating a response to an instruction. Given a criterion, we ask human annotators to give judgments on its utility. They indicate this on a scale of 1-3, where 1 that evaluating that criterion will have no impact on the response, whereas 3 means that it is extremely useful for knowing a quality of a response. We report this as an average score across the human annotation set.

**Obviousness ($O$):**    Given a criterion, we ask human annotators to evaluate if a criterion is "obviously" implied by the instruction or if it is not obvious. A criterion directly stated in the instruction is obvious, but criteria that are not explicitly stated in the instruction may or may not be obvious to the evaluator based on context or prior knowledge. We report this as an average score across the human annotation set.

We detail annotation subset, hiring process, and annotation instructions in Appendix E.

## 5 Experimental Setup

### 5.1 Dataset Curation

We evaluate criteria generation methods on a subset of instructions from BigGenBench (Kim et al., 2024a), ChatBotArena (Chiang et al., 2024), Art or Artifice (Chakrabarty et al., 2024a), Dolomites (Malaviya et al., 2025a), WritingBench (Wu et al., 2025), InfoBench (Qin et al., 2024), WildBench (Lin et al., 2024), and MT-Bench (Zheng et al., 2023). For all these datasets we keep instructions that have one of the keywords 'write', 'create' or 'develop'. We also filter for instructions that are in English, single-turn and focus on creative writing and planning tasks. Appendix A describes this data selection process in more detail.

| Dataset | # | Evaluation | Expert |
|---|---|---|---|
| BGB | 50 | Human | ✗ |
| Art or Artifice | 12 | Human | ✓ |
| Dolomites | 519 | Human | ✓ |
| InfoBench | 158 | Human | ✓ |
| MT Bench | 9 | LLM | ✗ |
| WildBench | 357 | LLM | ✗ |
| WritingBench | 235 | LLM | ✗ |
| Ask-then-Critique | 310 | Human | ✓ |
| ChatbotArena | 337 | - | - |

Table 1: All datasets considered in this work. We filter to a subset of instructions that require procedural, methodical or creative long-form outputs, discarding many examples from datasets like MT Bench.

To augment and diversify the set of instructions considered in this work, we collect a new dataset called **Ask-then-Critique**. This dataset consists of user-specified instructions, model responses from different LLMs (namely, ChatGPT, Gemini, or LLaMa), and user-written natural language critiques representing their evaluation. The goal of collecting Ask-then-Critique was to have users evaluate LLM responses for their own instructions to get more reliable critiques. This is different from other datasets considered in this work, where human-written criteria were not provided by the issuer of the instruction. Details about how we collected Ask-then-Critique are in Appendix A.2.

Table 1 enumerates the datasets along with the number of tasks used for this work. For most of these datasets, except ChatbotArena and MT-Bench, there are criteria that the dataset owners use for evaluation. Some of these criteria are human and expert written, where as the remaining datasets have LLM generated criteria that are human verified. We indicate this information in Table 1. Also, note that these dataset specific criteria do not have coverage of all evaluation properties of an instruction and hence can be variable in quality. We show examples of some of the instructions and constraints in Table 5.

### 5.2 Baselines

**Instruction Decomposition (ID)** generates criteria for **x** by decomposing the instruction into a list (Qin et al., 2024). For every instruction, we prompt an LLM to generate a list of instruction level criteria. This method targets capturing explicitly stated information in the instruction. **LLM-Prompted (LLM)** generates criteria by prompting an LLM for a set of evaluation criteria (Lin et al., 2024). Compared to Instruction Decomposition, this approach can go beyond criteria explicitly mentioned in the instruction. A related method, **LLM-Prompted to $n$ (LLM-$n$)** This baseline is stronger version of **LLM**, where we over generate the criteria and then subsequently rank them. This approach is not commonly used in the literature, but we explore it to take advantage of our proposed generate and rank pipeline. We find it to be particularly effective when combined with the EVALAGENT criteria.

| | # | $S \uparrow$ | $I \uparrow$ | $O$ | $U$ |
|---|---|---|---|---|---|
| ID | 6 | 0.13 | $0.50^{\dagger}$ | 0.96 | 2.94 |
| LLM | 9 | $0.10^{\dagger}$ | $0.67^{\dagger}$ | - | - |
| LLM-$n$ | 30 | $0.09^{\dagger}$ | $0.82^{\dagger}$ | 0.78 | 2.67 |
| EA-Web | 22 | **0.14** | **0.88** | 0.58 | 2.46 |
| Human | 9 | 0.21 | 0.81 | 0.73 | 2.69 |

Table 2: Automatic metrics comparing goodness of criteria generated by different methods. We see our proposed method has higher specificity and lower word-overlap with the instruction.$^{\dagger}$: signifies where the difference is significant with p $< 0.05$.

## 6 Results

### 6.1 Properties of generated criteria

**Criteria generated by EVALAGENT are more implicit.** Results in Table 2 show that EA-Web's criteria are specific and have less word overlap with the instructions, suggesting they are more implicit. Furthermore, we plot the distribution of the Specificity and Implicit scores in Figure 4, which shows that human-written and human-verified criteria have flatter distributions for both metrics. This flatter distribution of scores supports our hypothesis that humans expect a good response to satisfy implicit criteria. Criteria formulated by humans may also show lower direct overlap with the instruction due to the human distilling the main ideas into more generalized or abstract terms.

Table 11 in the appendix shows examples and scores for criteria for an instruction. We highlight words that lead to the specificity value for that criterion, such as the word "deprivation' in "*The response should focus on the emotional impact of sleep deprivation on relationships, work, and daily life.*"

**EVALAGENT's criteria are non-obvious** Table 2 reports the human judgment scores for Obviousness and Utility, where we find that EVALAGENT achieves a low obviousness score on average by our human annotators while maintaining a high utility score. *ID* generates the criteria with the highest utility; this is expected since the criteria are essentially constraints decomposed from the instruction itself and are necessary for instruction-following. LLM-prompted criteria are classified as mostly obvious, with a slightly lower utility score.

To dig deeper into these ratings, Table 15 in the appendix shows human judgments along with two tasks alongside criteria produced by LLM-*n* and our system. For utility, we report an average score from three annotator ratings, and for obviousness we show the number of annotators that found the criterion obvious. Note that obviousness is a subjective dimension and depends on the familiarity of the annotators with the instruction they are evaluating. For example, '*the response should include relevant emojis*' gets rated obvious by only one annotator out of three; similarly the criterion '*The response should create movement by transitioning from a specific moment to the broader implications of the characters*' is not obvious to any of the annotators, since it is very specific to creative writing and talks about narrative pacing. We show the average Utility and Obviousness scores from human annotators across datasets in Figures 8 and 9 of the Appendix.

**EVALAGENT generates highly actionable criteria** Table 3 reports actionability scores of various criteria. EA-Web achieves the

| Method | Initial | Edited | Δ | A |
|---|---|---|---|---|
| Claude-3.5-Sonnet | | | | |
| ID | 0.77 | 0.90 | 0.13 | 0.58 |
| LLM-*n* | 0.62 | 0.91 | 0.28 | 0.75 |
| EA-Web | 0.51 | 0.90 | **0.39** | **0.79** |
| Human | 0.73 | 0.91 | 0.18 | 0.66 |
| GPT-4o | | | | |
| ID | 0.79 | 0.90 | 0.10 | 0.52 |
| LLM-*n* | 0.65 | 0.92 | 0.27 | 0.77 |
| EA-Web | 0.57 | 0.90 | **0.34** | 0.77 |
| Human | 0.78 | 0.92 | 0.14 | 0.63 |
| Llama-3.1-8B-Instruct | | | | |
| ID | 0.76 | 0.88 | 0.13 | 0.53 |
| LLM-*n* | 0.60 | 0.90 | 0.30 | 0.75 |
| EA-Web | 0.47 | 0.86 | **0.40** | 0.74 |
| Human | 0.69 | 0.88 | 0.19 | 0.58 |

Table 3: We report Δ improvement in pass rate post-editing with criteria generated by different systems. We see that EA-Web generates the most actionable criteria, leading to the largest improvements. It is also able to find harder to satisfy criteria.

highest actionability score, indicated by the largest improvement (Δ) between the pass rates of edited and initial responses. Additionally, EA-Web has the lowest pass rates among all systems indicating that the system identifies more nuanced and challenging criteria. Furthermore, we find this lower pass rate is correlated with criteria rated as less obvious by humans; results can be seen in Figure 11 of the Appendix.

Table 13 enumerates initially unmet criteria in a response generated by GPT-4o and reports their corresponding outcomes after editing by the same model. Criteria generated by EA-Web are more implicit and seem useful for evaluating a response to the instruction. For

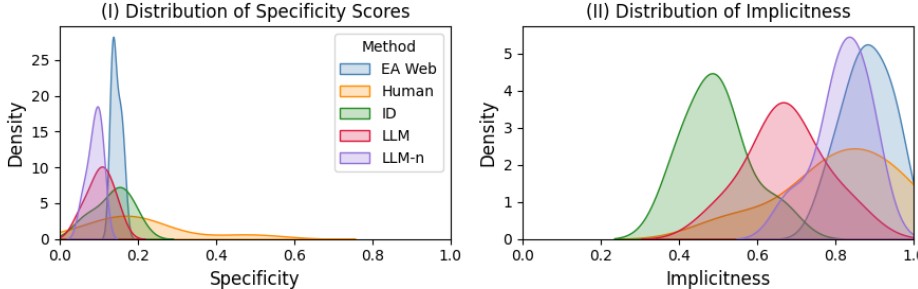

Figure 4: Distribution of the Specificity and Implicitness scores for each method. From (I) we see that EA-web generates a larger number of criteria with specificity higher than LLM-*n*. From (II) we see that EA-web generates criteria that has less overlap with the instruction compared to LLM-*n* and is more implicit.

|  | ID | LLM | LLM-30 | EA-Web | LLM-60 | LLM-30+30 | EA-Full |
|---|---|---|---|---|---|---|---|
| Art or Artifice | $0.04_{(6)}$ | $0.15_{(15)}$ | $0.47_{(30)}$ | $0.41_{(39)}$ | $0.54_{(60)}$ | $0.52_{(60)}$ | $\mathbf{0.62}_{(69)}$ |
| WritingBench | $0.52_{(9)}$ | $0.53_{(15)}$ | $0.74_{(30)}$ | $0.66_{(20)}$ | $0.82_{(59)}$ | $0.82_{(60)}$ | $\mathbf{0.86}_{(50)}$ |
| Ask-then-Critique | $0.29_{(4)}$ | $0.50_{(15)}$ | $0.64_{(30)}$ | $0.44_{(16)}$ | $0.71_{(60)}$ | $0.70_{(60)}$ | $\mathbf{0.74}_{(46)}$ |
| Average | $0.28_{(7)}$ | $0.39_{(15)}$ | $0.62_{(30)}$ | $0.50_{(25)}$ | $0.69_{(60)}$ | $0.68_{(60)}$ | $\mathbf{0.74}_{(55)}$ |

Table 4: We report recall values for human criteria across three datasets. We compare our proposed system, EA-Full against two LLM baselines. (1) LLM-60: we use LLM-*n* with n = 60 to match number of criteria generated by EA-Full (2) LLM-30+30: we prompt an LLM to generate additional 30 criteria to match our merge setup described in Section 3. EA-Full achieves a higher recall against human criteria than LLM baselines.

instance, in a financial analysis reporting task, LLM-generated criteria primarily focus on instruction level details. In contrast, EA-Web, generates criteria that describe key factors in a financial analysis report. Some EA-Web criteria which are not actionable are valid, but out of scope, for instance "*the data should be presented in the form of charts for better readability*". Models considered in this work cannot generate charts.

## 6.2 Alignment with human criteria

**Setup:** We now evaluate how well our generated criteria align with human-written criteria. Our proposed system, EA-Full (Section 3), combines EA-Web and LLM-*n*, resulting in a larger set of criteria that are then ranked.

We compare this approach against two LLM-based baselines. For fairness, we need these baselines to have access to the same number of criteria as EA-Full. For the first baseline we prompt LLM-*n* to generate the same total number of criteria as our system generates on average. We call this LLM-60 since we prompt the system to generate 60 criteria. The second baseline prompts the LLM to generate additional criteria beyond the initial set in EA-Full. This closely reflects our setup of combining EA-Web criteria with LLM-*n* where we add supplementary criteria to an initial set. We call this LLM-30+30 since we *add* another 30 criteria to an existing LLM-*n* list. Similar to EA-Full, we first generate criteria and then rank them as discussed in Section 3. We report these results on three datasets with varying types of human-written criteria: expert (Art or Artifice), human verified (WritingBench) and self-issued instruction criteria (Ask-then-Critique).

**Results:** From Table 4 we see that our system is able to capture human-criteria better than simply prompting an LLM to generate more criteria. Figure 5 further investigates the rate at which different systems recall human-annotated criteria, comparing our proposed system, EA-Full against LLM-60 and LLM-30+30. For WritingBench and Ask-then-Critique, all systems show similar performance until $k = 10$, after which EA-Full shows significant

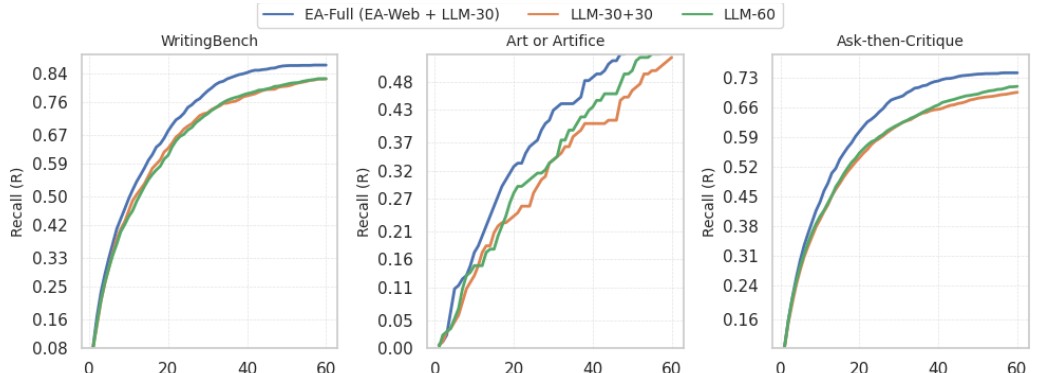

Figure 5: Recall@k for three datasets. For each method we calculate recall by prompting human criteria against the generated criteria (Section F.3). EA-Full retrieves human criteria at a higher rate compared to equivalent LLM baselines.

gains. This trend is even more pronounced for Art or Artifice, indicating that our approach captures nuanced criteria related to creative writing sooner compared to other baselines.

In Table 14 we show several criteria recalled by our system against the human criteria. We are able to match the human criteria, even though they vary in specificity. For example, the constraint *'the response should be complete and provide details about activities'* is entailed by *'the response should include outdoor activities'*, *'the response should include local cultural activities'* and *'the language should be descriptive'*. This highlights that we are able to capture criteria of a given human written constraint.

## 7 Related Work

**Generating Criteria**   Benchmarks like IFEval (Zhou et al., 2023) and COLLIE (Yao et al., 2024) add verifiable constraints to instructions, but these constraints rarely target deeper stylistic or content improvements critical for advancing LLM performance. InfoBench (Qin et al., 2024) only focuses on instruction decomposed constraints. WildBench (Lin et al., 2024) proposes a benchmark for more real-world queries evaluated via task-specific checklists. These checklists are LLM prompted and human-verified, but lack any grounding. VibeCheck (Dunlap et al., 2025) focuses on discovering criteria from a pair of preferences such that these criteria are able to differentiate between preferences. In contrast, our work centers on creating implicit yet grounded criteria that are adaptable across various feedback protocols and downstream applications.

**LLM-based Evaluators**   Recent research has explored the use of LLMs as evaluators for generative tasks, such as instruction following, in popular benchmarks like AlpacaEval (Li et al., 2023) and MT-Bench (Zheng et al., 2023), however these mostly focus on a predecided criteria or the general criteria of 'overall quality'. LLM-based evaluators are also applied in self-improvement tasks (Madaan et al., 2023; Yuan et al., 2024). Kim et al. (2024c) focuses on improving LLM-based scoring models. Our research complements these efforts by generating criteria suitable for integration within these systems.

## 8 Conclusion

We introduce a framework, EVALAGENT for generating implicit and grounded criteria for open-ended response evaluation. We do this by agentically searching the web for instructional documents for doing the task and synthesizing a task-specific criteria. Our evaluation shows that we can generated implicit, less obvious, and actionable criteria that capture what humans think is important. This framework paves the way for scalable evaluation of LLMs across a broad range of writing tasks.

**Limitations** EvalAgent relies on web-grounded retrieval to identify expert-written documents for criteria generation, which constrains it to the quality and scope of information available online. In niche domains, relevant content may be scarce or difficult to retrieve. For example, for the instruction *"write an article that talks about obtaining the necessary organic amount of a substance to make nanoparticles."*, EvalAgent generates a query *"how to write a clear procedure for synthesizing nanoparticles"*. This is a very niche domain and searching for the query either leads to scientific PDFs which are currently not extracted by our system, or leads to content related to nanoparticles instead of writing procedures, so we do not get any useful criteria from this query. However, as more information is indexed or as we expand our systems' abilities to process information in PDFs and other modalities, EvalAgent will become more effective, and the quality of what can be retrieved will continue to improve.

Moreover, adhering to the guidelines of experts on the web implicitly centers the perspectives of authors of retrieved documents in evaluation. In this study, these authors are all writing in English. Moreover, the views of these experts could be misaligned with the views of a different set of experts; e.g., if writing guidelines on the web encourage engagement via "clickbait" but an expert journalist would disagree with these. However, this effect was not a first-order problem we noticed in our analysis of our system's behavior, so we did not conduct further analysis of it. We note that our web page filtering did reduce the presence of overly generic criteria, but we did not observe this process to be amplifying any particular biases.

Finally, our work introduces additional overhead compared to directly prompting an LLM for evaluation criteria. Specifically, retrieving relevant web documents incurs latency, and multi-stage prompting with commercial APIs increases computational cost. However, these are one-time costs: both retrieval and criteria generation are conditioned solely on the instruction and can be precomputed prior to model evaluation. As such, they do not impact the runtime cost of benchmarking LLMs. Furthermore, the overall cost of the system is expected to decrease as inference costs for sufficiently capable LLMs become cheaper over time.

## Acknowledgments

This work was partially supported by NSF CAREER Awards IIS-2145280 and IIS-2145479, NSF grant IIS-2107524, NIH grant 1R01LM01460001, the NSF AI Institute for Foundations of Machine Learning (IFML), the NSF-Simons AI Institute for Cosmic Origins (CosmicAI) under NSF Cooperative Agreement 2421782 and Simons Foundation MPS-AI-00010515, and a grant from Open Philanthropy. Thanks to Kathryn Kazanas, Rohan Deshmukh and Benjamin Lew for the human annotations.

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

## A  Datasets

### A.1  Dataset Descriptions

We describe the datasets considered in this work along with instruction filtering we do to obtain the relevant subset. We also describe any human written or human verified criteria in the dataset.

**BigGenBench** (Kim et al., 2024a, BGB) is a benchmark with task specific fine-grained evaluation criteria. We only consider the subset of instructions that fall under the categories of theory of mind (*writing a speech*), instruction following (*education content creation*, *executable actions*) and planning (*constrained planning*, *travel planning*). This dataset has human written evaluation criteria for each instance which rates a response on a scale of 1-5. For each instance we use the provided Likert description for rating score 5.

**ChatbotArena** (Chiang et al., 2024) is a multilingual dataset with real queries. Since some instances in this dataset can be multi-turn, we only consider the first turn. We also only keep instances in English. For further filtering, we keep instructions with a positive creativity and real world tag. We also ensure that the subset of instances have one of the following keywords: 'write', 'develop', and 'create'. This dataset does not have any constraints or criteria provided.

**Art or Artifice** (Chakrabarty et al., 2024a) has creative writing instructions with expert-curated criteria. We consider all 12 instructions from this dataset. This work proposes

14 tests called Torrance Test of Creative Writing for evaluating creativity and short story writing. We consider these 14 as the criteria for the instructions in this dataset.

**Dolomites** (Malaviya et al., 2025a) is a long-form benchmark for evaluating LLMs on realistic domain-specific writing tasks. The dataset is composed of methodical tasks structured in the form of a task objective, procedure, input, and output. We use the task objective as the main instruction, and the procedure is rewritten into a criterion using GPT-4o to make it of the form *'the response should...'*.

**WritingBench** (Wu et al., 2025) is a designed to evaluate LLMs at multiple types of writing such as creative, persuasive, informative, and technical writing. These instructions are synthetically generated but manually verified. The task specific evaluation criteria are generated by prompting an LLM for format, length and content-specific constraints and is used for Likert scale evaluation on 1-10. We only keep instructions less than 400 words long. We rewrite the highest rating Likert description into a criterion of the form *'the response should...'* for our analysis.

**InfoBench** (Qin et al., 2024) is an instruction following benchmark. We filter instructions to have the following keywords: 'create', 'write' or 'develop'. We also filter out any math instructions. This dataset uses human written decomposed checklists as the evaluation criteria. We use the human written checklists as the criteria for this dataset.

**WildBench** (Lin et al., 2024) consists of human-chatbot conversation logs along with LLM-generated and human-verified evaluation checklists. We filter the dataset to keep the first turn of the conversation, instructions which are tagged to be 'creative writing' and 'planning' instructions. Similar to other datasets, we filter these to have the following keywords: 'write', 'create', and 'develop'.

**MT-Bench** (Zheng et al., 2023) is a benchmark containing 80 multi-turn hand designed questions. We only consider the writing subset of this dataset along with the instruction issued at the first turn. This dataset also has preferences between 6 model responses for all the instructions. To get constraints for this dataset, we prompt an LLM to give us a post-hoc reasoning for why a response was preferred over the other, and we convert this reasoning into a list of criteria.

### A.2 Ask-then-Critique

Our primary aim for collecting a new dataset was to have evaluations conducted by users who are also issuers of the instruction, so that the evaluations are genuine and the evaluators are invested in the problem/task they are evaluating. To collect this dataset, we added an opt-in study to one of the assignments of a graduate level NLP class. Instructions given to them are mentioned in box A.3.

The final dataset has 156 unique responses with 312 instructions and 624 evaluations. One thing to note here, human annotators provide their judgment in the form of natural language critiques. We rewrite these in this dataset to be formatted as criteria (*'the response should ..'*) for our experiments.

### A.3 Task categories

Datasets listed above cover a broad category of tasks, including financial reports to drafting emails. Figure 6 shows a high level distribution of these categories. We prompt `gpt-4o-mini` for a writing category output given an instruction and then cluster all of

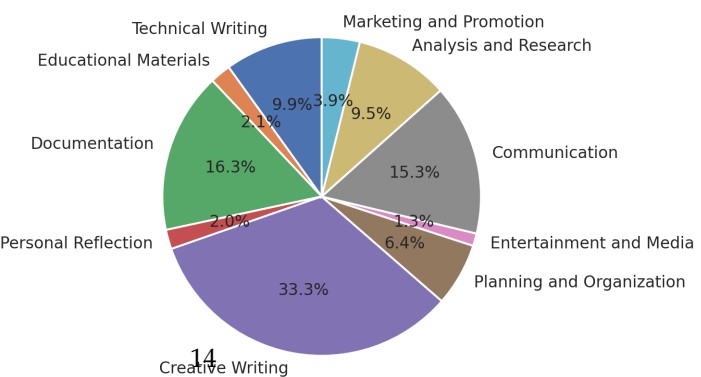

Figure 6: Distribution of task categories across datasets considered in this work

the outputs into high level task categories. We briefly describe what these categories entail:

**Creative Writing**: tasks like writing fiction stories, travel blogs, character sketches, academic essays, poetry, lyrics, humorous essays, comedy books, and radio scripts.

**Documentation**: tasks like synthesizing technical reports, legal documents, scientific reports, internal memos, restaurant menus, and HR reports.

**Communication**: tasks like writing emails, social media posts, tweets, reviews, letters, speeches, presentations, requests, alerts, and text messages.

**Technical Writing**: tasks like writing scientific protocols, fitness guides, game designs, self-help guides, and gardening guides.

**Analysis and Research**: includes different types of analysis like literary, investment, financial, academic, regulatory, and marketing.

---

**Ask-then-Critique Instructions**

We want you to explore use cases around AI assistants. You can opt-in for your responses to this part to be included as part of a research study; more details will follow. You will be investigating how LLMs perform and evaluating them across two realistic tasks. You will give LLMs instructions to do various tasks that you might use AI assistants for. For evaluation, we want you to analyze the aspects you use to see if the LLM response satisfied you or not. An aspect describes a dimension, such as factuality, structure, safety, logic, flow and more.

For instance, you might prompt an AI Assistant with the following: "Write me a 5-day travel itinerary for Austin, Texas around Austin City Limits. This is for a blog post." Some of the aspects I might use for evaluation can be: (1) Correctness: the response correctly talks about things in Austin, Texas; (2) Engagingness: the response is engaging, has personal anecdotes and persuades the user to visit Austin, Texas; (3) Coherence: the response is well structured with good topical grouping of things

Your task: Please come up with two different prompts which might be useful to you and which also emphasize a balance of structural, tone and content constraints. Your prompts should follow the following constraints:

- They should require a longer form answer (at least 1 paragraph) and shouldn't simply be knowledge based (e.g.: when was Abraham Lincoln born?)

- They should be something that you can evaluate for utility (e.g., writing a literature review, writing a tweet thread for your paper based on abstract and conclusion, writing an essay on a topic etc). We want you to be able to critique these prompts.

- You should be comfortable critiquing the output to the question (don't ask about quantum physics if you don't know about it)

Then, get an answer from two of the following models: ChatGPT (www.chatgpt.com), Meta AI (www.meta.ai), or Gemini (https://gemini.google.com/). You may use any version of these models you like, either free or paid. Finally, evaluate the response. Your evaluation should be done by you and not use AI assistants. You should evaluate at least 3 aspects.

For each response, what you submit should look like:

Prompt: [prompt]

Response (including model name): [response]

[Aspect name: your assessment and reasoning]

E.g., "Engagingness: the response is engaging and the personal anecdotes persuade me to visit Austin"

You will evaluate a total of 4 responses (2 prompts x 2 models) in this format with at least 3 aspects each. Each (prompt, model) combination can use distinct aspects, so you could have 12 (or more), or as few as 3 distinct aspects. You should find a flaw in at least 2 of the responses. To achieve this, you can use one of the following strategies: (1) Ask about more obscure knowledge; (2) Place more constraints on the response; (3) Ask for more interesting output formats (a LinkedIn post, a TED talk, etc.).

Ideally your prompts should be as creative and distinct as you can make them: if something comes to mind from your work or a particular life situation, that's great. However, please don't add any personal identifiable information that you are not comfortable with the course staff potentially seeing as they grade the assignments.

**Educational Materials**: includes planning and writing tasks like creating tutorials, curriculum, syllabus, quizzes, rubrics, and flashcards.

**Entertainment and Media**: includes planning and writing tasks like educational podcasts, promotional videos, and music playlists.

**Marketing and Promotion**: includes writing tasks like business proposals, advertising slogans, and product pitches.

**Personal Reflection**: includes writing tasks like personal advice, reflection blogs, social media bios, parenting advice, and pet care blogs.

# B EvalAgent Parameters and Prompts

In this section we describe each step and parameters of our proposed method along with the corresponding prompts.

**Step 1: Query Generation** The first step of our method is query decomposition. Using the prompt given below, we formulate queries that will help retrieve instructional URLs. We generate 2-3 queries per instruction.

> **Prompt B.1: Query Generation**
>
> Instruction: {{ Instruction }}
> What should I google so I learn useful advice on writing a great response to the above instruction? Give a JSON response with three most important google queries as keys. The queries should be in the form of "how to" or "what are" etc and the value is what we want to learn from the query. The queries should focus on seeking actionable writing advice instead of focusing on abstract knowledge concepts.

**Step 2: Retrieving Expert Advice** For each query generated in Step 1, we run the query through the Google Search API and retrieve 30 URLs. Each URL content is rated based on two metrics we define in Section B.1: (1) absolute rating on a scale of 1-5 for the goodness and perceived expertise of the content (prompt B.8) (2) absolute rating on a scale of 0-2 for the relevance of the content to the query and instruction (prompt B.9). We sort the URLs by the content relevance and scored expertise and choose the top-5. The selected URLs have an average URL rating of 3.77 (on a scale of 1-5) and an average relevance rating of 1.48 (on a scale of 0-2). We show examples of instructions, along with the queries and corresponding URLs and text snippets Table 16.

Once we have the filtered URLs, we pass them through the document-grounded QA prompt (listed below) and get $r_k$ for each query.

> **Prompt B.2: Document Grounded Query Answering**
>
> ### Article: {{ article }}
> ─────
> Answer the following question from the article above: {{ query }}
> The question is to help me write an answer to the following instruction: {{ instruction }}
>
> ONLY answer the question and not the instruction.
> I am looking for nuanced advice to the question above, include any obscure or minute points like structuring the content, the tone, highlighting important things etc.
> If the article does not have any useful advice, then return "no answer".

At the end of this step, we combine content from all URLs using the summary prompt listed below to get a list of summaries for each query.

> **Prompt B.3: Summarizing URL Content for a Query**
>
> ## Instruction: {{ instruction }}
>
> ## In order to evaluate a response to the above instruction, I looked up different expert advice on {{ query }}. I found the following two responses:
> ### Expert Advice 1:
> {{ advice_1 }}
>
> ### Expert Advice 2:

{{ advice_2 }}

## Your goal is to identify new information in Advice 2 compared to Advice 1. Once you have identified that information, integrate it in Advice List 1 and return a combined list of points. Do not delete anything from Advice 1.
Include any unique and nuanced details from Advice 2, eg: any thoughts on how the structure should be, tone etc. If there are examples of what phrasing etc to use please include them under any relevant points. You MUST NOT draw any inferences on your own. DO NOT answer the instruction. You only choose to merge or combine details as needed.

**Step 3: Criteria Generator** The final step in our proposed method combines all criteria lists generated for each query in Step 2 and aggregates that information. The response then goes through a re-writing and filtering prompts to ensure alignment to the instruction and removal of any irrelevant criteria.

We use the following prompt for combining all query-specific criteria. We do this two-criteria lists at a time.

```
Prompt B.4: Aggregating Query Content
```

### Advice list 1: {{ list1 }}

### Advice list 2: {{ list1 }}

## Add any new information and details from List 2 to List 1. Don't delete anything from List 1.
## Include any unique details from List 2, eg: any thoughts on how the structure should be, tone etc
## If there are examples, please include them under any relevant points
## Summarize the information into one list of points, each point should be unique and structured such that it reflects an evaluation criterion.

We use the following prompt to re-write the instructional web-document criteria to be a checklist and in the form of criteria that can be used for evaluation.

```
Prompt B.5: Re-writing Set
```

### Instruction: {{ instruction }}

### Tips/Suggestions: {{ tips }}

—————
Convert the tips to constraints/conditions so they can be used for evaluating a response to the given instruction. Eg: "the response should.."
Constraints should: - **Be answerable by 'yes' or 'no'**, with 'yes' meaning that the response successfully met the corresponding requirement.
- **Be comprehensive, but concise**, meaning that all criteria directly relevant to the instruction should be represented by a question, but only questions that are very clearly relevant should be included.
- **Be precise**, meaning that constraint should avoid vague wording and evaluate specific aspects of a response, directly using the phrasing of the instruction where appropriate.
Give a new line separated numbered list, where each line is one constraint. Ensure they are rephrased to align with the instruction.

After re-writing we run the criteria list through a filtering step that removes any out of scope or irrelevant criteria.

```
Prompt B.6: Filter
```

## Instruction:
{{ instruction }}

## Criteria for evaluating a response to the above instruction:

{{ criteria }}

_________

For the above criteria, which of them can be used to evaluate a response to the instruction?
Note: the response is not interactive, we cannot evaluate if the response went through iterative revisions and there is no feedback process. If there is any criteria evaluating for these, then mark them no.
Think step by step and give a response for each criteria point, end the reasoning with "therefore, the criteria applies yes" or "therefore, the criteria does not apply no". Return a JSON where the key is the criteria number and the value is the reasoning for whether or not the criteria applies.

**Rank:** In this step we re-order the generated criteria from above based on the relevance of a criterion to the instruction. We prompt an LLM for this rating on a scale of 0-10.

---

**Prompt B.7: Relevance Ranking**

How relevant is the following aspect to evaluating a text only response to the following instruction? Note, that we cannot use these for evaluating iterative editing etc. so any aspect which mentions that should be rated low.
Instruction: {{ instruction }}

Aspect: {{ aspect }}

Think about it step by step and return a number between 0 and 10. End your response with 'therefore, the score is:'. 0 is the lowest relevance and 10 is the most relevant/useful aspect.

---

## B.1 URL Filtering

As mentioned in the section above, we filter the retrieved URLs based on two criteria (a) the goodness of the URL content (b) the relevance of the content to the instruction. We list the prompts used for this rating below.

---

**Prompt B.8: URL Content Rating**

## URL: {{ url }}

## Content in the URL: {{ content }}

________________

Rate the goodness of the above URL and content based on the following criteria: Score: 1 - Very Poor
Expertise: Content appears unprofessional or amateurish, with little evidence of expertise. Reliability: The domain is untrustworthy or unknown, such as suspicious .com sites, clickbait domains, or personal blogs with no credentials. Clarity: Information is vague, confusing, or contradictory, with no actionable advice. Marketing: The content is overtly promotional, focused primarily on selling a product or service rather than providing useful information.
Score: 2 - Poor
Expertise: Some effort to appear knowledgeable, but lacks depth or substantiation (e.g., no citations, shallow explanations). Reliability: Domain is not widely recognized, or the platform has a mixed reputation. Clarity: Advice is somewhat unclear or too generic to be useful. Marketing: There is a noticeable marketing agenda, but it does not entirely overshadow the content.
Score: 3 - Fair
Expertise: Content demonstrates moderate expertise, though it may lack nuance or depth. Basic credibility is evident. Reliability: The domain is moderately reputable (e.g., a well-known .com site, or an .org/.edu/.gov site with minor credibility concerns). Clarity: Information is reasonably clear and somewhat actionable, though not highly detailed or specific. Marketing: Some promotional elements are present but do not dominate the content.
Score: 4 - Good
Expertise: Content is well-researched and written by someone with clear expertise or authority in the field. Reliability: The domain is highly reputable, such as a trusted .edu, .gov, or widely

---

respected .com/.org site. Clarity: Advice is clear, specific, and actionable, with practical steps or examples. Marketing: Minimal promotional content; the focus is on providing value to the reader.
Score: 5 - Excellent
Expertise: Content reflects deep expertise, with authoritative writing, detailed explanations, and citations or references to credible sources. Reliability: The domain is extremely trustworthy, such as a government, academic, or highly esteemed organization. Clarity: Advice is exceptionally clear, highly specific, and immediately actionable, tailored to the reader's needs. Marketing: No marketing agenda is evident, or if present, it is subtle and does not detract from the quality of the information.
Only return the score number at the end.

---

**Prompt B.9: Instruction, URL Content Rating**

## URL: {{ url }}

## Title: {{ title }}

## Article: {{ article }}

_______________________________________

Does the web content provide any advice that will help me write a response to the following instruction:
Instruction: {{ instruction }}

2 - yes and it provides specific advice that will help me write a response to the instruction
1 - yes but it only provides general advice that will help me write a response to the instruction
0 - no, it doesn't help at all
Only give the numerical response and description.

---

# C  Computational Cost

The cost for running EvalAgent and its corresponding components are given in Table 6. We report the input and output tokens per call for each of the steps in our proposed work, along with the total number of tokens and associated costs. We see that the bulk of the cost is the filtering step during retrieval.

# D  Baseline Methods

In this section we describe the prompts used with the baseline methods.

---

**Prompt D.1: Instruction Decomposition**

### Instruction: '{ instruction }'
Decompose the instruction into the constraints that the instruction explicitly mentions. Then convert the constraints into a checklist. The checklist response should have each new line as one numbered point. Each point should start with "the response should". Do not mention any constraint that is not explicitly in the instruction.

---

**Prompt D.2: LLM**

Help judge an AI assistant's response to an instruction by providing an evaluation checklist.
### Instruction: '{ instruction }'
## Task Details Your task is to come up with an evaluation checklist list for a given Instruction. This evaluation checklist should be a list of questions that ask whether or not specific criteria relevant to the instruction were met by an AI assistant's response.
Criteria covered by your checklist could be explicitly stated in the instruction, or be generally sensible criteria for the problem domain. You should, however, try to be concise and not include unnecessary entries in your checklist.
Checklist questions should:

- **Be answerable by 'yes' or 'no'**, with 'yes' meaning that the response successfully met the corresponding requirement.
- **Be comprehensive, but concise**, meaning that all criteria directly relevant to the instruction should be represented by a question, but only questions that are very clearly relevant should be included.
- **Be precise**, meaning that checklist questions should avoid vague wording and evaluate specific aspects of a response, directly using the phrasing of the instruction where appropriate.
Give a list where each line corresponds to one factor. The factors should start with 'the response should'.

```
Prompt D.3: LLM-n
```

Help judge an AI assistant's response to an instruction by providing an evaluation checklist.
### Instruction: '{ instruction }'.
## Task Details Your task is to come up with an evaluation checklist list for a given Instruction. This evaluation checklist should be a list of questions that ask whether or not specific criteria relevant to the instruction were met by an AI assistant's response. Criteria covered by your checklist could be explicitly stated in the instruction, or be generally sensible criteria for the problem domain. You should, however, try to be concise and not include unnecessary entries in your checklist. Checklist questions should: - **Be answerable by 'yes' or 'no'**, with 'yes' meaning that the response successfully met the corresponding requirement. - **Be comprehensive, but concise**, meaning that all criteria directly relevant to the instruction should be represented by a question, but only questions that are very clearly relevant should be included. - **Be precise**, meaning that checklist questions should avoid vague wording and evaluate specific aspects of a response, directly using the phrasing of the instruction where appropriate.
Give a list where each line corresponds to one factor. The factors should start with 'the response should'. Return 30 factors.

# E   Human Evaluation

To assess Utility and Obviousness of the generated criteria, we hire two annotators on Upwork and one annotator who has a bachelors degree in linguistics. Our Upwork qualifications included a success rate of at least 90% and native or bilingual speaker of English. We paid a base rate of $30/hour.

Each annotator went through a qualification round post which they were given the main task. In total, the annotators gave judgments on 700 criteria across datasets and systems with 54 unique instructions. Table 8 shows the total number of criteria sampled across datasets and systems.

> **Annotation Instructions**
>
> **Context**: We are broadly interested in evaluating LLM responses to an instruction. Step 1 towards this research is to first define and identify criteria that would be useful for evaluating such responses. We built a system that generates instruction specific criteria for evaluating LLM responses. We would like your help to evaluate these criteria!
>
> Goal for this annotation study: Given an instruction and a criterion, we want to understand the quality of this criterion and its relevance in evaluating an LLM response to the instruction. You will be given: (1) An Instruction: A prompt for generating an LLM response (2) A Criterion: A criterion for assessing the quality of the LLM response.
>
> Your Task: You will rate the evaluation criterion along two dimensions: **Obviousness and Relevance.**
>
> **1. Obviousness (Yes or No ):** Does the evaluation criterion clearly follow from the instruction?
>
> **0 (Not Obvious):** The criterion is not explicitly mentioned in the instruction. It is also not obvious.
>
> **1 (Obvious):** The criterion is either explicitly stated in the instruction or is common sense for assessing the response.
>
> Example:
> *"The response should have correct grammar"* is a criterion that might not be explicitly stated but is fairly obvious - Yes, obvious
> *"The response should have bolded text for highlights"* might not be stated in the instruction but is not obvious - No, not obvious
>
> **2. Relevance (Scale: 1 to 3)** How important is the criterion for assessing an LLM generated response?
>
> **1 (Not Relevant):** The criterion has little to no connection to the instruction, so it won't be relevant to evaluating an LLM response to the instruction
>
> **2 (Somewhat Relevant):** The criterion is "nice-to-have" but if it's not met by the LLM response, it would still not significantly impact LLM response quality
>
> **3 (Highly Relevant):** The criterion is essential - if not met, the LLM response quality might be noticeably affected.

### E.1 Annotator Agreement

Table 7 shows the agreement between the annotators on utility and obviousness broken down by method. We use Fleiss Kappa to calculate agreement. Lower agreement between annotators is expected by systems that generate implicit criteria, as what is obvious to one annotator may not be obvious to another (i.e. some may be experts in a topic while others may not). Similarly, for utility, one annotator may find implicit criteria to be very impactful while others may find it to be only slightly useful or a "nice to have". Because of this subjectivity in annotation, we report the average rating of obviousness and utility in our metrics over majority vote.

### E.2 Average Utility and Obviousness Scores

We report the average annotator utility and obviousness scores in Figure 7. We break these averages down further by method and dataset in Figure 8 and Figure 9. While Figure 8 shows comparable Utility, the Obviousness rating from the annotators differs, showing that EVALAGENT consistently finds less obvious criteria with high utility.

| Method | O | U |
|---|---|---|
| ID | 0.88 | 0.64 |
| LLM-$n$ | 0.23 | 0.25 |
| EvalAgent | 0.25 | 0.26 |
| Human | 0.26 | 0.29 |
| Average | 0.34 | 0.34 |

Table 7: Average annotator agreement for Obviousness and Utility broken down across methods.

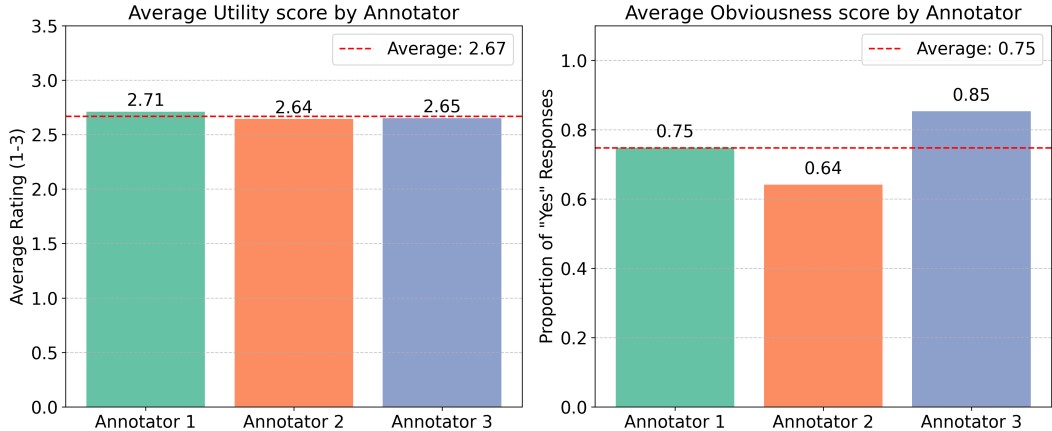

Figure 7: Average scores for Utility and Obviousness from annotators.

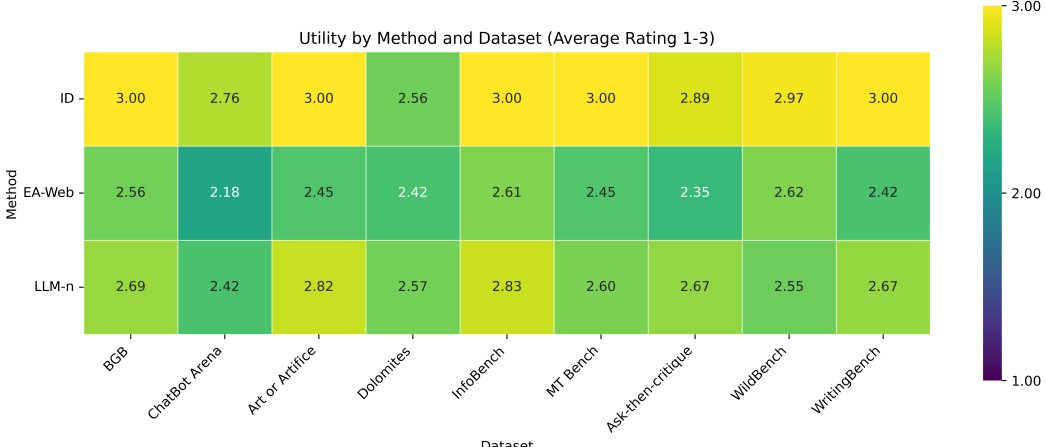

Figure 8: Average of Utility from the human annotators across methods and datasets

## F Automatic Metrics

### F.1 Lexical metrics: Specificity and Word Overlap

**Specificity** $S$: We calculate specificity of a criterion by taking a maximum of the normalized inverse word frequency of the words in the criterion. Given criterion $c$, we calculate NIWF as the following:

$$\mathbf{NIWF_c} = \max_{w \in \mathbf{c}} \left( \frac{\log(1 + |\mathcal{R}|)}{f_w} \right).$$

(1)

This is reasonable since a criterion is specific as long as it contains some specific words. We calculate the frequency of a word in the en-

| Dataset | ID | LLM-n | EA-Web | Human | # Ins |
|---|---|---|---|---|---|
| BGB | 23 | 36 | 36 | 8 | 8 |
| ChatBotArena | 7 | 15 | 15 | - | 3 |
| Art or Artificie | 16 | 25 | 28 | 13 | 6 |
| Dolomites | 12 | 21 | 20 | 23 | 5 |
| InfoBench | 22 | 35 | 25 | 24 | 9 |
| MT Bench | 12 | 19 | 20 | 15 | 4 |
| Ask-then-Critique | 15 | 27 | 20 | 19 | 6 |
| WildBench | 10 | 14 | 14 | 10 | 3 |
| WritingBench | 27 | 46 | 41 | 38 | 10 |
| Total | 144 | 238 | 219 | 150 | 54 |

Table 8: Distribution of criteria and instructions across datasets and systems for the human annotation task.

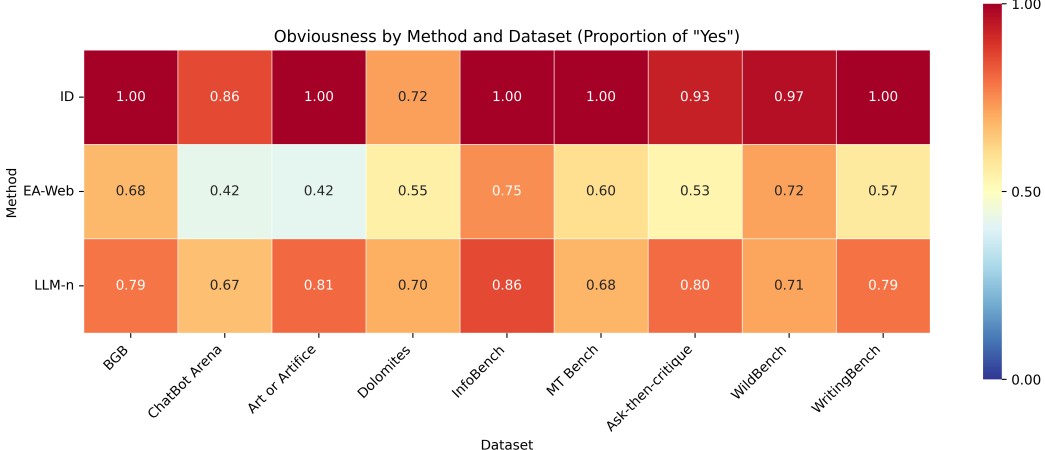

Figure 9: Average scores of Obviousness from the human annotators across methods and datasets.

tire corpus of criteria generated across datasets and systems. This is done to ensure that the specificity is being calculated against the same pool. We report the specificity scores for all datasets in Table 9.

We also look at the self-specificity of the method, i.e., specificity of a criterion generated by a method with respect to a pool of criteria generated by the same method. We show the distribution of scores in Figure 10. From this figure we see that *ID* tends to be the most specific, when considering its own pool of criteria. This is intuitive since the instruction level constraints are generally unique to the instruction (unless there are common things shared across instructions like '*this should be a New Yorker style story*'). For *LLM*, when we only generate the natural distribution of criteria (without over generating), we see that it has a higher specificity as well. This is because without over generating, LLM tends to give you more instruction level and explicit criteria, which causes it to be more specific. For LLM-*n*, we see that specificity scores are lower than EA-Web. This probably because over generating the criteria will give some that are shared across instructions. For example, for a lot of structured writing tasks like TED Talks or Blogs it could be '*the response should be have a clear introduction, body and conclusion*', which lowers the specificity of criteria.

**Implicitness** $I$   To calculate the implicitness of a criterion **c**, we look at the word overlap ($WO$) of the criterion with the prompt **x** and report 1-$WO$. The ratio is calculate as following:

$$\text{WO}(x,c) = \frac{|\text{W}(x) \cap \text{W}(c)|}{|\text{W}(c)| + \epsilon} \qquad (2)$$

Where $W(p)$ is a set of non-stopword tokens from the lowercase $p$.

We report the implicitness scores for all datasets in Table 10.

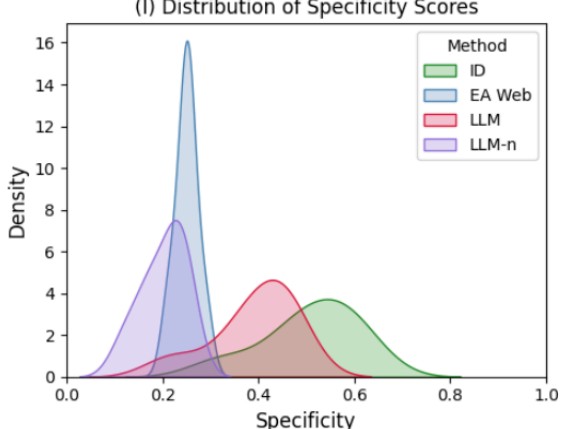

Figure 10: We look at the self-specificity of each method, i.e. specificity of a criterion generated by a method with respect to a pool of criteria generated by the same method.

### F.2   Actionability

We define actionability of a criterion as:

$$\Delta_c = \mathbb{1}\left[f(x, G(x,y,c), c) = 1\right] - \mathbb{1}\left[f(x,y,c) = 1\right]$$

where $f$ is a criteria-based scoring which gives binary judgments. $\mathbf{x}$ is the prompt, $\mathbf{y}$ is an initial response generated by $G$. When given a criterion $c$, $G(x, y, c)$ edits $y$ to try and satisfy $c$.

**Data split:** To conduct this experiment we sample 200 instructions across datasets considered in this task. Note, we leave out ChatBotArena, MT-Bench from this analysis. Both are preference datasets with no human written criteria to compare with.

**Models** We calculate actionability with respect to three models - GPT-4o-2024-11-20, Claude-3-5-sonnet-20241022 and Llama-3.1-8B-Instruct. The model used for generating the initial response is the same as the model used for editing.

**Satisfaction prompt:** We use the following prompt to evaluate if a criterion is satisfied by a response. This prompt returns binary judgment.

```
Prompt F.1: Criteria-based scoring prompt
```

You are a teacher, evaluate a student's response to an instruction.
A student is given the following instruction to answer:
Instruction: {{ instruction }}

Student's Response: {{ response }}

Does the response satisfy the following criteria: {{ criterion }}

Think step by step and end your thought with 'therefore, the answer is yes' or 'therefore, the answer is no'.

| Dataset | # instructions |
|---|---|
| BGB | 34 |
| Art or Artifice | 12 |
| Dolomites | 26 |
| InfoBench | 34 |
| WildBench | 32 |
| WritingBench | 34 |
| Ask-then-Retrieve | 31 |

Table 12: Number of instructions considered for the the actionability experiment in Section 6.

**Editing prompt:** We use the following prompt to edit model responses that do not satisfy a criterion.

```
Prompt F.2: Editing Prompt
```

Instruction: {{ instruction }}

Response: {{ response }}

## The above response to the instruction does not satisfy the following constraint: {{ criterion }}

## Edit the above response such that it satisfies the above constraint. Do not include a preamble.

**Actionability vs Obviousness:** We test how actionability is related to obviousness of generated criteria. We consider the human annotated subset for this experiment. First, we generate initial responses using GPT-4o-2024-11-20 and then calculate an initial pass rate with criteria generated from different systems. We use gpt-4o-mini-2024-07-18 for evaluation. We then plot this against the human annotated obviousness judgments.

Figure 11 plots how the initial fail-rate of criteria (criteria that were generated and not satisfied in the initial responses) varies with the human evaluations of obviousness per criteria. We find that EVALAGENT consistently generates criteria that is less obvious and less likely to be initially satisfied indicating challenging and useful criteria.

### F.3 Recall

We evaluate how frequently human-generated or verified criteria for a given instruction are represented in the generated criteria. To calculate this we use the following prompt:

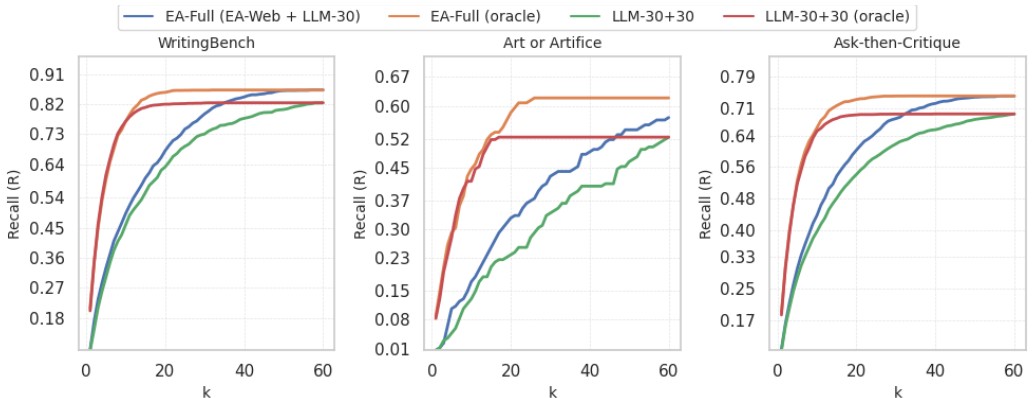

Figure 12: We report the oracle results for Recall@k across three datasets. There is a big gap between the relevance ranking we use vs the oracle. We leave the development of a better scoring function to future work.

```
Prompt F.3: Recall Prompt
```

## I have an existing list of evaluation criteria. I want to compare a new criterion to this list. The lists are given below:
Existing criteria: {{ existing_criteria }}
New criterion: {{ new_criterion }}

## Determine if the new criterion is already mentioned in the existing list. You can generalize the new criterion to see if it aligns with the existing list or vice-versa Return a JSON in the following form:
{
"copy of a point from the general point above":"is the specific criterion entailed in the existing list or not"
}

For a prompt **x**, if there are $H$ human-criteria, we run all $h \in H$ through the prompt above and get one decision for the prompt **x**. We then calculate recall@k for all instances ($N$) as following:

$$\text{Recall@k} = \frac{1}{N} \sum_{i=1}^{N} \frac{|H_i \text{ in the entailed list}|}{|H_i|} \quad (3)$$

**Recall@k with oracle:** Figure 12 shows also the oracle performance of these systems. This shows that there is a significant gap between the ranking method and the oracle. We leave improving the ranking of criteria for future work.

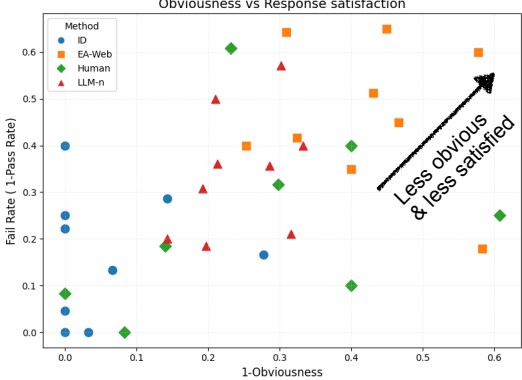

Figure 11: Trend of the percentage of unsatisfied criteria by how unobvious they are across methods and datasets.

**Dataset**: BGB
**Instruction**: Design a travel plan for a tourist traveling to the given destination.
The tourist has a list of requirements and you should design your plan such that it satisfies
all of these requirements.
Destination: Paris
Requirements:
- Total Duration: 2 days and 1 night
- Transportation: Walk
- Must Have: Eiffel Tower, Louvre Museum, Escargot
- Optional: Croissant, Onion Soup, Notre Dame Cathedral
**Constraint**:The response presents a well-thought-out, efficient, and realistic itinerary that includes
all must-have experiences within the walking-only constraint and incorporates all optional items,
demonstrating excellent planning and optimization for an enriching tourist experience.

**Dataset**: Art or Artifice
**Instruction**: Write a New Yorker style fiction given the plot below.
Make sure it is atleast 1500 words.
Directly start with the story, do not say things like 'Here's the story [...]:'

Plot:
An observer becomes entranced by a seemingly ordinary couple on the street, follows them home,
and then watches them from outside in the rising floodwaters, drawing an eerie connection between
the woman and a discarded, burned chair they'd noticed earlier.

**Constraint**: The response should have effective narrative pacing controlling the perceived speed and
rhythm at which a story unfolds

**Dataset**: Dolomites
**Instruction**: Draft a comprehensive legal framework for the use of blockchain technology in the
creation and transfer of digital assets within the real estate industry.
**Constraint**:It should include a precise review of pertinent real estate laws and regulations, identifying
any recent changes or updates.

**Dataset**:WritingBench
**Instruction**:Please help me write a review of the TV drama "The Long Night" (Silent Truth).
**Constraint**:The response should have meaningful insights into the complexity of the plot, including
narrative arcs and story development.

**Dataset**: InfoBench
**Instruction**: Generate a non-disclosure agreement of two pages (each page is limited to 250 words)
for a software development project involving Party A and Party B.
The confidentiality duration should be 5 years.
The first page should include definitions for
key terms such as 'confidential information', 'disclosure', and 'recipient'.
On the second page, provide clauses detailing the protocol
for the return or destruction of confidential information,
exceptions to maintaining confidentiality, and
the repercussions following a breach of the agreement.
Please indicate the separation between the first and second pages
with a full line of dashed lines ('——').
Also, make sure that each page is clearly labeled with its respective page number.
**Constraint**:Does the second page of the generated non-disclosure agreement provide clauses detailing
the protocol for the return or destruction of confidential information?

**Dataset**:WildBench
**Instruction**:Create a training program that can be done at home without any equipment, and without
a pullup bar. It must be heavily focused at muscle hypertrophy and strength gain. Training days
are 6 times a week, and one extra day is the rest day. Don't add any cardio, and include ab and core
exercises in the daily program instead of reserving a specific day for those exercises. Every single
muscle in the body must be trained at least twice a week, with maximum focus towards gaining
muscle.
**Constraint**: Are the exercises suitable for being performed at home without any equipment or a pullup
bar?

Table 5: Examples of instructions and corresponding dataset constraints from the datasets
datasets considered in this work.

| EA-Web Cost | Num calls | Input tokens (per call) | Output tokens (per call) | Total cost (with GPT-4o-Mini) |
|---|---|---|---|---|
| (1) Query Generation | 1 | 113 | 94 | $0.00007 |
| (2) Expert Retrieval | | | | |
| Filter (for three queries) | 33 | 7705 | 10 | $0.04 |
| | 28 | 3990 | 16 | $0.02 |
| | 31 | 5410 | 9 | $0.03 |
| Query-Answering | 15 | 5446 | 249 | $0.01 |
| Summarization | 5 | 1211 | 795 | $0.00 |
| (3) Aggregation + Criteria Generation | 4 | 1367 | 723 | $0.003 |
| Total | 117 | 627009 | 11761 | $0.10 |
| Total (without filtering) | 25 | 93331 | 10691 | $0.02 |

Table 6: Computational costs for running EvalAgent broken down by three steps: (1) Query Generation; (2) Expert Retrieval; (3) Criteria Generation. For Step (2) we show the step broken down by different prompt types used.

| Dataset | ID | LLM | LLM-n | EA-Web | Human |
|---|---|---|---|---|---|
| BGB | 0.17 | 0.11 | 0.09 | 0.16 | 0.48 |
| Art or Artifice | 0.15 | 0.15 | 0.1 | 0.15 | 0.06 |
| Dolomites | 0.06 | 0.05 | 0.06 | 0.16 | 0.19 |
| InfoBench | 0.09 | 0.08 | 0.07 | 0.13 | 0.13 |
| Ask-then-Critique | 0.13 | 0.09 | 0.09 | 0.13 | 0.22 |
| WildBench | 0.19 | 0.13 | 0.11 | 0.14 | 0.2 |
| WritingBench | 0.15 | 0.11 | 0.11 | 0.13 | 0.16 |
| Average | 0.13 | 0.10 | 0.09 | 0.14 | 0.21 |

Table 9: Specificity ($S$) values for criteria generated by different methods for datasets considered in this work.

| Dataset | ID | LLM | LLM-n | EA-Web | Human |
|---|---|---|---|---|---|
| BGB | 0.48 | 0.67 | 0.81 | 0.87 | 0.79 |
| Art or Artifice | 0.5 | 0.67 | 0.86 | 0.96 | 0.98 |
| Dolomites | 0.65 | 0.84 | 0.89 | 0.95 | 0.93 |
| InfoBench | 0.41 | 0.62 | 0.8 | 0.83 | 0.54 |
| Ask-then-Critique | 0.52 | 0.74 | 0.87 | 0.9 | 0.89 |
| WildBench | 0.51 | 0.67 | 0.8 | 0.87 | 0.73 |
| WritingBench | 0.4 | 0.51 | 0.69 | 0.78 | 0.81 |
| Average | 0.50 | 0.67 | 0.82 | 0.88 | 0.81 |

Table 10: Implicitness ($I$) values for criteria generated by different methods for datasets considered in this work.

| (**Ask-then-Critique**)**Instruction**:I want to give a short podcast on why sleep is important to the well-being of an individual. However, my audience is those who rarely get sleep. How can I go about this? | | | |
|---|---|---|---|

| System | Criterion | $S$ | $I$ |
|---|---|---|---|
| Human | Response should have depth in discussing the importance of sleep, providing detailed ==talking== points, examples, and relevant reasons. ==sleep==, acknowledging their feelings and experiences. | 0.07 | 1.0 |
| | Response should have empathy, acknowledging and understanding the feelings and struggles of individuals who ==rarely== get sleep. | 0.1 | 0.7 |
| LLM | The response should suggest ways to incorporate humor or ==light-heartedness==, if appropriate. | 0.5 | 0.83 |
| | The response should include a summary of key points to ==reinforce== the message. | 0.005 | 1.0 |
| EA-Web | The response should focus on the emotional impact of sleep ==deprivation== on relationships, work, and daily life. | 0.5 | 0.9 |
| | The response should provide practical ==takeaways== or actionable advice after sharing personal stories. | 0.009 | 1.0 |

Table 11: Examples of criteria generated by different systems along with the Specificity ($S$) and Implicitness ($I$) values for an instruction from the Dolomites dataset. We ==highlight== the word in each criterion with the highest specificity.

| (**WritingBench**) **Instruction**: Please write an analysis report on the revenue of wealth management products for China Merchants Bank and China Construction Bank (with a length of 3000-4000 words), based on the following annual report data of wealth management products from 2018-2022 for both banks: #Bank 1 Annual Details #Bank 2 Annual Details | | | |
|---|---|---|---|

| System | Criterion | Actionable | Implicit |
|---|---|---|---|
| LLM-$n$ | The response should analyze the net profit for both banks from 2018 to 2022. | ✓ | ✗ |
| | The response should provide a comparison of the total assets of both banks in 2022. | ✓ | ✗ |
| EA-Web | The response should contain a concise executive summary that captures the key findings and recommendations regarding revenue trends, profitability, and significant changes over the years. | ✓ | ✓ |
| | The response should consider different scenarios that could impact future performance, discussing potential risks and opportunities. | ✓ | ✓ |

| (**Ask-then-Critique**) **Instruction**: I am getting a new 8-week old golden retriever puppy. Please document the materials I will have to gather and things I need to prepare for this new adventure. Additionally, please give me specific tips for raising a golden retriever. I want a comprehensive guide for owning a new puppy as a first-time dog owner. Thanks! | | | |
|---|---|---|---|
| LLM-$n$ | The response should highlight the benefits of adopting from a shelter or rescue. | ✓ | ✗ |
| | The response should address the financial responsibilities of owning a puppy. | ✗ | ✓ |
| EA-Web | The response should suggest a daily schedule that includes feeding, potty breaks, exercise, and playtime. | ✓ | ✓ |
| | The response should explain the grooming needs specific to golden retrievers and the importance of regular grooming. | ✓ | ✓ |

Table 13: Examples of criteria generated by LLM-$n$ and EA-Web that were initially unmet by a GPT-4o response. We see that LLM-$n$ generate instructions that less implicit and more grounded in the instruction.

| Human criteria | EA-Web Generated criteria |
| --- | --- |
| Response should provide a logical flow that addresses the client's issues, the proposed solution, and the expected ROI. | The response should emphasize the return on investment (ROI) for the client. |
| | The response should have a clear structure, including sections for introduction, needs assessment, proposed solution, benefits, implementation steps, and Q&A. |
| Response should be mindful of time constraints, ensuring that all points are relevant and succinct to fit within a typical onehour meeting. | The response should be concise and clear, avoiding overwhelming the audience with information. |
| Response should not include overly lengthy paragraphs but rather concise points that can be easily articulated. | The response should be formatted in a way that is easy to follow |
| | The response should be concise and clear, avoiding overwhelming the audience with information. |
| Response should have a clear structure that is easy to read and organized, preferably in a bulleted format | The response should provide a structured itinerary divided into clear sections for each day of the weekend trip |
| Response should be complete and provide descriptive details about activities in each area. | The response should include activities or attractions to visit in Boston. |
| | The response should include outdoor activities to enjoy the fall foliage. The response should highlight activities and excursions that immerse visitors in the local culture. The response should use engaging and sensory-rich descriptions to bring local attractions to life. |
| Response should have a fluid structure that feels personal and heartfelt, rather than a list format. | The response should include a personal touch, such as a memorable experience or lesson learned |
| | The response should mention any lessons learned during the author's time at the organization |

Table 14: Examples of human written criteria in the Ask-then-Critique dataset that matched with a corresponding EA-Web generated criteria and had no match in LLM-$n$. Our system captures criteria that are important for humans when evaluating responses to their instruction.

| Method | Criterion | $U$ | #$O$ |
|---|---|---|---|
| **Instruction** :Act as an meta worker write me good caption for my reel which is based on fun dance good vibes great company | | | |
| LLM-$n$ | The response should not be overly complex or convoluted. | 2.67 | 3 |
| | The response should reflect the personality of the creator or brand. | 2.67 | 2 |
| | The response should include relevant emojis to enhance the visual appeal of the caption. | 2.00 | 1 |
| EA-Web | The response should incorporate humor or a light-hearted tone in the caption. | 2.67 | 2 |
| **Instruction:** Craft an intriguing opening paragraph for a fictional short story. The story should involve a character who wakes up one morning to find that they can time travel. | | | |
| LLM-$n$ | The response should align with the theme of time travel. | 3.00 | 3 |
| | The response should avoid ambiguity regarding the character's ability. | 2.33 | 2 |
| EA-Web | The response should create movement by transitioning from a specific moment to the broader implications of the character's newfound ability to time travel. | 2.00 | 0 |
| | The response should subtly hint at the challenges or adventures that lie ahead for the character. | 2.00 | 1 |

Table 15: Examples of human annotations for two instructions and their corresponding criteria generated by LLM and EA-Web. We show human evaluation scores: Avg Utility ($U$) across annotators and the number of annotations that marked the criterion obviousness (#$O$).

**(WritingBench) Instruction**: Prepare a plan for a show in an art gallery.
**Query**: How to plan a art exhibition
**Aspect** the response should create a detailed timeline for the exhibition planning process
**URL:** https://www.contemporaryartissue.com/how-to-organize-a-successful-art-exhibition
**Snippet:** Develop a timeline and budget. A solid timeline and budget are essential for smooth execution. Break down tasks and allocate resources for: Space rental, Artwork materials and framing, Marketing (digital and physical), Digital tools (e.g., virtual tours or AR components), Factor in sustainable practices, such as renting materials, using recycled supplies, or partnering with eco-conscious vendors, to minimize your environmental footprint. Include a contingency fund for unexpected expenses to avoid last-minute stress.

**(Art or Artifice) Instruction**: Write a New Yorker style fiction given the plot below. Make sure it is atleast 1500 words. Directly start with the story, do not say things like 'Here's the story [...]:'Plot: A solitary man walking in a remote mountainous region comes across a car crash, and stays by the side of the lifeless female victim, narrating stories of his past and reflecting on the impermanence of events and life itself, while awaiting emergency services amidst the looming presence of wilderness.

**Query**: how to build atmosphere and setting in writing
**Aspect**: the narrative should juxtapose memories of life with the stark reality of death to emphasize the impermanence of life.
**URL**: https://www.thesaurus.com/e/writing/creating-atmosphere-mood/
**Snippet:** In this non-fiction travelogue, David Foster Wallace is talking about his experiences on luxury cruises. He opens by placing the reader directly onto a cruise ship. In the span of a paragraph, the reader experiences awe, curiosity, amusement, disgust, wonder, and excitement. Yet Wallace uses formal language ("I have seen") and repetition (there's that anaphora for you) to ironic effect. This creates an interesting juxtaposition of the elements of a tall tale with a bit of anthropological distance. This example, in particular, shows how mood can function independently from the atmosphere, and how both can change abruptly with the use of language.

**(Ask-then-Critique) Instruction**: Write a TED talk about how to resume your career in the United States after a long pause from professional work due to familial responsibilities. Explain how I had been in a similar situation and how to deal with its mental and financial implications while providing resources to do so. Include placeholders where I can include personal details as needed.

**Query**: what are key elements of a compelling TED talk
**Aspect**: the response should include a personal anecdote about resuming a career
**URL**:https://sixminutes.dlugan.com/how-to-deliver-talk-life/
**Snippet**: Tell a story that hasn't been told before As a journalist I had an advantage. I'm a professional storyteller. Yet I still had to find a new story, a story about being homeless that hadn't been told before. So I told my story. It's easy to hide behind talking about other people in similar situations, with similar issues, but the powerful story, the one people want to hear, is your story.

Table 16: Examples of instructions, queries generated along with the evaluation aspect and it's grounding URL with a snippet.

