# OpenReview forum: "EvalAgents: Discovering Implicit Evaluation Criteria from the Web"
_colmweb.org/COLM/2025/Conference — COLM 2025_

### Official Review · Reviewer_1Txy · 2025-04-18

**Rating:** 7
**Confidence:** 5
**Ethics Flag:** 1

**Summary:**

This paper presents a highly interesting and significant subtask in the context of LLM-as-a-Judge, namely the analysis of the effectiveness of evaluation criteria. The authors propose an LLM-prompting-based eval-agent framework that enhances the reliability of LLM-as-a-Judge by generating more useful evaluation criteria through a retrieval-augmented generation procedure. Furthermore, they introduce several metrics for assessing the quality of evaluation criteria, which provides valuable insights for future research. Extensive experimental results demonstrate the superior reliability of the proposed eval-agent compared to LLM-based prompting baselines.

**Questions To Authors:**

None

**Reasons To Accept:**

​1. The research problem addressed in this paper​ represents an underexplored yet critically important direction in the field of ​LLM-as-a-Judge. The study’s motivation is well-justified and holds significant implications for advancing this research area.
​2. This work introduces a novel RAG-based framework (evalagent)​​ for constructing ​evaluation criteria, along with ​multiple dimensions and metrics​ for assessing their quality.

**Reasons To Reject:**

### **Weaknesses / Limitations:**
1. **Insufficient Literature Review**
   The paper lacks a thorough discussion of prior work in **LLM-based criteria generation**, which has been preliminarily explored in **critique and LLM-as-a-Judge research**. For instance:
   - **MultiCritique [1]** proposes a **hierarchical taxonomy** for generating criteria from **coarse-grained to fine-grained** levels.
   - **HD-Eval [2]** trains a high-quality criteria generation model using **human-annotated data**.
   A **comparative analysis** of these works is necessary to clarify the **novelty and contribution** of the proposed **evalagent** framework.

2. **Computational Overhead of EvalAgent**
   The framework involves **query generation, retrieval+summarization, and criteria generation**, which likely incurs **significant computational costs**. However, the paper lacks a **cost-efficiency comparison** between **evalagent** and baseline methods. A **runtime/memory analysis** should be included to provide a **more balanced evaluation** of the framework’s **practical applicability**.

3. **Heuristic Evaluation Metrics for Criteria Quality**
   The proposed **evaluation metrics** for criteria quality appear **somewhat heuristic**. The authors could **enhance reliability** by:
   - **Indirectly assessing criteria quality** via **critique or refinement performance** (e.g., **RealCritic [3]** evaluates critiques based on **revision quality**).
   - Clarifying the **pass rate (improvement) metric** (Section 6.1, Lines 318-325), which currently lacks **detailed methodology** (e.g., how improvement is quantified).

> [1] Training Language Model to Critique with Multi-agent Feedback
>
> [2] HD-Eval: Aligning Large Language Model Evaluators Through Hierarchical Criteria Decomposition
>
> [3] RealCritic: Towards Effectiveness-Driven Evaluation of Language Model Critiques

### **Minor Issues (Typos & Formatting):**
1. **Figure 5** contains a **LaTeX compilation error** (needs correction).
2. **Line 195**: "We report **S and W** across" → should be "We report **S and I** across" (likely a typo).
3. **Figure 3 (wrapfigure)** is **misaligned**, encroaching on the next section’s text (requires formatting adjustment).

---

> ### Author Response · Authors · 2025-06-02
>
> Thank you for your feedback! We can address the typos and formatting errors in the paper release. Addressing some of the limitations below:
>
> > Insufficient literature review and criteria quality
>
> We can add detailed differences between work that the reviewer suggested and our work in the future release. But highlighting some of the them:
>
> 1. While HD-Eval does criteria generation and pruning, the domain they operate in is very constrained compared to the creative and long form writing tasks that our paper focuses on. Their method also relies on human annotations for doing the preference guided aggregation, which are hard to obtain at scale. As a result, their system is not widely applicable like ours is and cannot be directly compared to. Moreover, they focus on correlations with human scores downstream which is not our main focus. We do believe that their pruning technique could potentially be adopted on top of the criteria generated by our system.
>
> 2. The RealCritic work for criteria revision could be added on top of the criteria we generate, but we view this as orthogonal to our contribution.
>
> 3. MultiCritique uses multiple LLM for generating feedback and proposes methods for fine-tuning better critique models. They do not test for specificity, implicitness and actionability of the criteria generated. Moreover, even with multiple LLMs the system is only utilizing parametric knowledge compared to the generated criteria based on expert advice as proposed by us.
>
> > Computational Overhead of EvalAgent
>
> The cost for running EvalAgent and its corresponding components is the following:
>
> | EA-Web Cost                       | num calls | input tokens | output tokens | cost (with GPT-4o-Mini) |
> | --------------------------------- | --------- | ------------ | ------------- | ----------------------- |
> | Query Generation                  | 1         | 113          | 94            | $0.000073               |
> | Expert Retrieval                  |           |              |               |                         |
> | Filter (for three queries)        | 33        | 254276       | 341           | $0.038346               |
> |                                   | 28        | 111707       | 437           | $0.017018               |
> |                                   | 31        | 167695       | 292           | $0.025329               |
> | Query-Answering                   | 15        | 81695        | 3728          | $0.014491               |
> | Summarization                     | 5         | 6057         | 3977          | $0.003295               |
> | Aggregation + Criteria Generation | 4         | 5466         | 2892          | $0.002555               |
> | Total                             | 117       | 627,009      | 11,761        | $0.1                    |
> | Total (without filtering)         | 25        | 93331        | 10691         | $0.02                   |
>
> As you can see the bulk of the computation/cost is the filtering step. But we want to emphasize that the cost of this system will go down as inference costs become cheaper.
>
> Compared to an LLM baseline, this is a higher cost of computation, but there are many benefits of this approach. Firstly, this is only a one-time cost since the retrieval and criteria generation only depends on the instruction, so a user can generate these one-time and use these to benchmark models. Secondly, the benefit of this style of evaluation is high, especially when the performance of LLMs on LLM generated criteria is saturated.
>
> We can include this analysis in any future version of the paper.
>
> > Clarifying the pass rate (improvement) metric
>
> Pass rate is the percentage of samples where the LLM response satisfies the corresponding evaluation criterion. We consider this in two settings (a) calculating pass rate for an “initial response” to the instruction from an LLM (b) for responses that do not satisfy their corresponding criteria, we refine them with that criterion as the feedback and re-calculate the pass-rate. We report this difference as the pass-rate improvement (with the same sample set) and actionability (over the initially unsatisfied samples). We also detail this process in Appendix E.2 but can make it more clear in any future version.

---

> > ### Comment · Reviewer_1Txy · 2025-06-04
> >
> > Thank you for your response. I will keep my score as 7.

---

### Official Review · Reviewer_393x · 2025-05-10

**Rating:** 5
**Confidence:** 4
**Ethics Flag:** 1

**Summary:**

This paper proposes EVALAGENT, a framework for automatically generating implicit, task-specific evaluation criteria for open-ended writing prompts. Instead of relying solely on LLMs or human-defined standards, EVALAGENT mimics how humans seek advice by querying the web, retrieving expert-authored instructional content, and synthesizing actionable evaluation criteria. The framework shows improved specificity, implicitness, and actionability over baseline methods across various writing tasks.

**Questions To Authors:**

1. Does the EVALAGENT framework depend on live internet access during inference? Or are documents retrieved and cached beforehand? How does this compare to the baseline LLMs' capabilities?
2. Were any of the baseline LLMs given access to external data (e.g., via tools, browsing)? If not, does that put them at an inherent disadvantage compared to EVALAGENT?
3. How do you ensure that the content retrieved from the web is factual, trustworthy, and not misleading? Are there mechanisms in place for hallucination or misinformation filtering?
4. How well does EVALAGENT generalize across domains where high-quality instructional material might be sparse (e.g., niche or emerging fields)?

**Reasons To Accept:**

1. The paper introduces a new method for discovering implicit evaluation criteria by grounding them in expert web content, rather than relying solely on LLMs or human annotators.
2. The multi-step framework, from query generation to ranking, closely reflects how humans gather instructional advice.
3. Extensive experiments across multiple datasets show consistent improvements in criteria quality, including actionability and alignment with human preferences.

**Reasons To Reject:**

1. The framework depends on web content, but the paper doesn’t deeply address safeguards against unreliable or low-quality sources. It's uncertain how well the system performs in areas with limited or sparse instructional content online.
2. It’s not specified whether models used during evaluation had real-time internet access or relied on cached data, raising reproducibility concerns.
3. Baseline LLMs may not have had access to the same external information, which could bias comparisons in favor of the proposed method.

---

> ### Author Response · Authors · 2025-06-02
>
> Thank you for your comments and feedback! Addressing your concerns and questions below:
>
> > Does the EVALAGENT framework depend on live internet access during inference? …. How does this compare to the baseline LLMs' capabilities?
>
> In general, yes, EvalAgent depends on live internet. But during our experiments, we first generated queries and retrieved corresponding documents, which were then cached. All subsequent experiments were performed on top of the cached results. This approach ensured reproducibility and consistency across experiments, as they are stable with respect to changes in the available web content.
>
> A baseline LLM does not require live Internet access, though in practice many users run queries via web APIs rather than on-device, and so have access anyway. In Tables 2, 3 and 4, we compare against baseline approaches which use a similar framework but without web retrieval. Our results show that the criteria grounded in web documents is more specific, less obvious and actionable compared to that generated by a baseline LLM. This shows that retrieval of instructional documents played an important role in the quality of criteria generated.
>
> > Were any of the baseline LLMs given access to external data (e.g., via tools, browsing)?  If not, does that put them at an inherent disadvantage compared to EVALAGENT?
>
> No, the baseline methods we compared against did not have access to external data or tools such as web browsing during criteria generation. Our goal is to compare with approaches from prior work, which do not use web access in this way. This also enables us to compare how parametric criteria (which is how the models are used currently) compare with EvalAgent.
>
> In general, EvalAgent could be implemented with a retrieval-augmented LLM, and we think this would be great to explore! However, closed-source retrieval-augmented LLMs are not transparent about their queries and thus we do not believe an implementation based on this achieves the required scientific reproducibility for our system.
>
> > How do you ensure that the content retrieved from the web is factual, trustworthy, and not misleading? Are there mechanisms in place for hallucination or misinformation filtering?
>
> In Appendix B.1 we elaborate on the URL filtering procedure used. We filter out any advertising looking documents and documents where the content appears as if it is not written by experts. In our current implementation we used an LLM to do this filtering. Unfortunately these checks only evaluate for overall quality of the instructional document rather than any external checks for factuality or trustworthiness. However, we anecdotally did not observe non-factual or misinformation sources to be heavily retrieved, as commercial search engines already attempt to exclude these. Also, we only retrieve the top-30 URLs and only keep 5 after filtering instead of retrieving a longer tail. More generally, our system could be extended with adding specialized filtering for identifying misinformation or non trustworthy web pages or human-in-the-loop validation of retrieved pages, but we leave the development of such a system to future research.
>
> > How well does EVALAGENT generalize across domains where high-quality instructional material might be sparse (e.g., niche or emerging fields)?
>
> We agree that web-grounding in EvalAgent is only as good as the information on the web. For example, for the instruction:  “write an article that talks about obtaining the necessary organic amount of a substance to make nanoparticles.”, one of the generated queries is:  “how to write a clear procedure for synthesizing nanoparticles”.  This is a very niche domain, so searching for these either leads to scientific PDFs which we currently do not parse, or leads to content related to nanoparticles instead of writing procedures, so we don’t get any criteria from this query. However, as more information is indexed or as we expand our systems’ abilities to process information in PDFs, EvalAgent will become more effective and the quality of what can be retrieved will continue to improve!

---

> > ### Comment · Reviewer_393x · 2025-06-06
> > **response**
> >
> > Thank you for the detailed response.
> >
> > I would like to seek further clarification regarding your answer to the second question. You mentioned that “EvalAgent could be implemented with a retrieval-augmented LLM.” For step 1, I understand that the proposed method aligns with general RAG approaches. However, the distinction seems to lie primarily in steps 2 and 3. From my reading, it appears that these steps involve prompting the LLM to perform the task directly. I would appreciate it if you could elaborate further on how these steps differ from standard prompting approaches or clarify if there are additional novel aspects I may have overlooked.

---

> > > ### Author Response · Authors · 2025-06-06
> > >
> > > Thank you for your follow-up question!
> > >
> > > We should clarify our comment slightly. When we say that EvalAgent could be implemented with a retrieval-augmented LM (RALM), what we mean is that _part of_ the approach could use such an LLM. For instance, step 2 (corresponding to lines 4-10 of algorithm 1) could possibly be implemented with an RALM if prompted appropriately: this step takes a query, executes a search, filters the results, and aggregates answers to the queries from the retrieved results. In Step 3, we aggregate the summaries across queries and then re-write them to be actionable, precise and aligned to the instruction. This ensures that _multiple aspects_ of the writing task are covered by the evaluation criteria and all aspects are grounded in expert instructions.
> > >
> > > There are two reasons why Step 2 and Step 3 are implemented with separate systems and prompts. First, EvalAgent’s performance does depend on the queries for retrieval of high-quality results. Replacing Step 2 with an off-the-shelf RALM may result in suboptimal retrieval (due to poor queries being generated) or poor filtering of the results (due to not having the separate filtering stage). Second, we observed that the aggregation pipeline across steps 2 and 3 preserved nuanced and long-tail criteria, which were missed if we combined these steps and directly generated criteria from the retrieved content, which would be a more standard RAG approach. That is, these steps need to be addressed in separate calls and then merged together to capture all the nuances of the task.
> > >
> > > So the difference from standard prompting arises precisely because we found differences that were needed to address the challenges of our problem setting. In theory, a very strong LLM could go from retrieved results to the final criteria in a single prompt, or a very strong RALM could do all of that and the retrieval. But this was not possible with existing LLMs, and our system’s design helps preserve long-tail criteria.

---

> > > > ### Comment · Reviewer_393x · 2025-06-09
> > > > **responese**
> > > >
> > > > Thank you for the prompt response. However, I still find limited differences between the proposed method and standard RAG, aside from applying it to different tasks. I will keep my score.

---

### Official Review · Reviewer_BmkZ · 2025-05-13

**Rating:** 7
**Confidence:** 3
**Ethics Flag:** 1

**Summary:**

LLM outputs are usually evaluated a set of criteria presented to human evaluators or other LLMs (e.g., LLM-as-judge settings). Strong answers typically rely on unstated criteria and conventions. The author(s) of this paper propose EvalAgent, a framework designed to automatically discover latent task-specific criteria. EvalAgent consists of 3 stages: a query generator, an expert retriever, and a criteria generator. They find that the criteria generated by EvalAgent are more implicit and specific, more actionable, and align better with human-written criteria.

**Questions To Authors:**

* There is a broken reference in Figure 5 (says ``Prompt ??'').

**Reasons To Accept:**

* There is a clear, agentic structure. The three stages---query generator, expert retriever, and criteria generator---are modular and can be swapped out with different LLMs as needed. This makes it applicable for both general and specialized LLMs.
* The evaluation section is broad and consists of a new dataset (Ask-then-Critique). Evaluation is across both automatic metrics and human evaluation. The improvements are notable, and human alignment improves without obviousness.
* The use of an expert retriever grounds the generated criteria in real-world expertise found on the web.

**Reasons To Reject:**

* The web-grounding is only as good as what is available on the web. For certain tasks or tasks in emerging domains, there may not be enough expert information online to generate a good set of criteria. It would be helpful to see when and how the EvalAgent pipeline fails.
* The interrater agreement, measured through Fleiss' kappa, is quite low for EvalAgent (0.25 for obviousness, 0.26 for utility). This implies that there is very high disagreement with the criteria generated using EvalAgent.
* There is no discussion about the potential amplification of biases found in online "expert" contexts. Relatedly, with the datasets, the Ask-then-Critique was collected using a graduate NLP class, and consists of evaluations and critiques from a highly educated and non-representative group of people completing the task.
* This reads as a very computationally intensive process; no details are provided about the computational costs of EvalAgent.

---

> ### Author Response · Authors · 2025-06-02
>
> Thank you for your feedback! We will fix the broken reference in the future versions of this paper.
>
> Addressing some of the concerns below:
>
> > The web-grounding is only as good as what is available on the web.
>
> We agree, although we would add that the quality of what can be retrieved will continue to improve. For example, for the instruction:  “write an article that talks about obtaining the necessary organic amount of a substance to make nanoparticles.”, one of the generated queries is:  “how to write a clear procedure for synthesizing nanoparticles”.  This is a very niche domain, so searching for these either leads to scientific PDFs which we currently do not parse, or leads to content related to nanoparticles instead of writing procedures, so we don’t get any criteria from this query. However, as more information is indexed or as we expand our systems’ abilities to process information in PDFs, EvalAgent will become more effective.
>
> > The interrater agreement, measured through Fleiss' kappa, is quite low for EvalAgent
>
> We explain the reason for this disagreement in Section D.1. First, many tasks considered here are inherently subjective (we give some examples in Table 12, like drafting a caption for an instagram reel or writing an opening paragraph for a fictional story) and we expect some level of disagreement. Secondly, with implicit criteria, it is harder to establish agreement between annotators since what is obvious to one annotator might not be obvious to another (i.e. some may be experts in a topic or have familiarity with certain domains while others may not).  We highlight some examples in Table 12. For the prompt that discusses writing a caption for a reel, the ratings will depend on the annotator's familiarity with social media.
>
> > There is no discussion about the potential amplification of biases found in online "expert" contexts.
>
> We aren’t quite sure what kinds of biases you’re referring to exactly. Any evaluation requires some prescriptive definition of what makes a good output. We agree that in principle, criteria scraped from “experts” the web could be misaligned with the views of a different set of “experts”; e.g., if writing guidelines on the web encourage engagement via “clickbait” but an expert journalist would disagree with these. However, this effect was not a first-order problem we noticed in our analysis of our system’s behavior, so we did not conduct further analysis of it. We note that our web page filtering did reduce the presence of overly generic criteria, but we didn’t observe this process to be amplifying any particular biases.
>
> > This reads as a very computationally intensive process; no details are provided about the computational costs of EvalAgent.
>
> The cost for running EvalAgent and its corresponding components is the following:
>
> | EA-Web Cost                       | num calls | input tokens | output tokens | cost (with GPT-4o-Mini) |
> | --------------------------------- | --------- | ------------ | ------------- | ----------------------- |
> | Query Generation                  | 1         | 113          | 94            | $0.000073               |
> | Expert Retrieval                  |           |              |               |                         |
> | Filter (for three queries)        | 33        | 254276       | 341           | $0.038346               |
> |                                   | 28        | 111707       | 437           | $0.017018               |
> |                                   | 31        | 167695       | 292           | $0.025329               |
> | Query-Answering                   | 15        | 81695        | 3728          | $0.014491               |
> | Summarization                     | 5         | 6057         | 3977          | $0.003295               |
> | Aggregation + Criteria Generation | 4         | 5466         | 2892          | $0.002555               |
> | Total                             | 117       | 627,009      | 11,761        | $0.1                    |
> | Total (without filtering)         | 25        | 93331        | 10691         | $0.02                   |
>
> As you can see the bulk of the computation/cost is the filtering step. But we want to emphasize that the cost of this system will go down as inference costs become cheaper.
>
> Compared to an LLM baseline, this is a higher cost of computation, but there are many benefits of this approach. Firstly, this is only a one-time cost since the retrieval and criteria generation only depends on the instruction, so a user can generate these one-time and use these to benchmark models. Secondly, the benefit of this style of evaluation is high, especially when the performance of LLMs on LLM generated criteria is saturated.
>
> We can include this analysis in any future version of the paper!

---

> > ### Comment · Reviewer_BmkZ · 2025-06-04
> >
> > Thank you for your detailed response. In particular, thanks for the detailed response to my question about the computational costs of the process. It would be great to see this in the paper.
> >
> > Thank you for clarifying that there is a filtering process. That said, I still believe that potential biases still warrants a brief discussion in this paper. What I mean by biases here is not the low-quality or overly generic information, but to systematic patterns that may arise from relying on a narrow distribution of "expert" advice (e.g., North American, English language, academically oriented content) or from a demographically homogeneous group of evaluators (e.g., the education backgrounds of graduate students in an NLP class). It would be helpful to see this at least acknowledged as a potential issue in discussions of limitations.
> >
> > I will keep my score as a 7.

---

### Official Review · Reviewer_tKpV · 2025-05-26

**Rating:** 5
**Confidence:** 3
**Ethics Flag:** 1

**Summary:**

The paper proposes a pipeline-based approach that identifies implicit evaluation criteria for structured writing tasks. This is done in a three-step sequence: (1) generation of queries that retrieve instructional Web documents which contain evaluation criteria, (2) a filtering and summarization step that narrows down what has been retrieved in the first step, and (3) a generation step that takes the output of the second step and turns that into evaluation criteria. Each step in the pipeline is done via prompt engineering accessing a LLM. Subsets of existing benchmarks are used to evaluate the work adopting technical measures as well as crowd-based assessments.

**Questions To Authors:**

* The choice of LLM is not really motivated. What is the reason for the choice? How do the findings compare to a different LLM?
* The reference to significance testing in Table 2 is not quite clear. The difference is significant compared to what? Also, do you apply a suitable correction such as Bonferroni?
* Would it not make sense to include another baseline that is not entirely based on LLMs but based on a manual creation of evaluation criteria?

**Reasons To Accept:**

* The paper is topically an excellent fit for COLM. It will be of interest to a fair number of attendees and is likely to stir some discussion.
* The authors provide an extensive appendix explaining every detail of the approach which helps making sure that the work is fully reproducible which is a major strength (however, the downside is that the extensive appendix makes the paper less self-contained than it could be as the reader needs to refer to information in the appendix to fully follow the discussion so that the paper reads more like a 15-page paper than a 9-page paper).

**Reasons To Reject:**

* The biggest weakness in my view is the lack of an extrinsic evaluation of the work. The authors argue that they “leave the exploration of downstream applications to future work” but without such evaluation the contribution is a lot less compelling than it could be, in particular as there is no evidence that the proposed methodology will beat plausible baselines.
* In part the work feels more like an elegant engineering solution than a contribution that offers substantial new insights. A contributing factor is the choice of a number of ad hoc settings (such as “google queries”, “30 results”, “above 2”, “above 0”,  “top-5” …)
* The paper would benefit from a Limitations section. While not strictly a requirement it does become clear that a number of experimental choices need to be critically discussed in the light of generalisability and other possible limitations.

Minor issue:

*  “Specificity” is perhaps slightly confusing as a term as it is commonly understood to refer a different concept in the context of evaluation (true negative rate).

---

> ### Author Response · Authors · 2025-06-02
>
> Thank you for your comments and feedback! Addressing some of the questions raised below:
>
>
> > The biggest weakness in my view is the lack of an extrinsic evaluation of the work
>
> We conduct a study of the actionability of our criteria (shown in Table 3 and Figure 10) which we argue is somewhat extrinsic. We show that various models have a lower pass-rate on criteria generated by our system, which is also correlated with criteria being non-obvious. We also show that LLMs can revise outputs to satisfy these criteria. This experiment suggests a possible application of our system: identify some criteria along which to improve a response and then improve it along those criteria.
>
> Such an application stops short of a fully automated pipeline for generally making responses “better” in a universal sense. For instance, we do not evaluate generic human ratings of responses and show that evaluation under our criteria correlates with these ratings. We believe that this goal is fundamentally very difficult to achieve. In a non-specialized domain such as blog writing, different humans apply different standards. LLMs already generate passable responses; they can be improved, but we argue that this involves tailoring the response improvement to a particular user’s desires. Imposing standards on evaluators (e.g., tell them what factors to look for) brings bias into the evaluation.
>
> Doing this work properly would require significant interface design and a human study that we see as a separate contribution unto itself. We view our work as providing a lever to surface criteria, but connecting that all the way to a downstream evaluation is beyond the scope of a single paper.
>
>
> >  In part the work feels more like an elegant engineering solution than a contribution that offers substantial new insights
>
> While our system does involve a substantial amount of engineering, we believe there is also a substantial intellectual contribution, namely our retrieval-based evaluation protocol and the empirical insights it enables. Our work takes a different perspective on the source and role of criteria than past work, which we believe meaningfully advances the literature in this area. Furthermore, our experiments did not show the system to be fragile with respect to hyperparameter choices, so we believe this contribution stands on its own independent of the system. We can add a detailed explanation of our hyperparameter choices to the appendix.
>
>
> > The paper would benefit from a Limitations section
>
> We can definitely add a limitations section to the paper, thanks for this!  A few points we think would be useful to add are:
>
> 1. Computational costs:  Compared to just using an LLM, retrieval involves extra latency and cost from using commercial APIs. However, because evaluation can be done on a relatively small number of instances for a given LLM, we do not believe this is a major limitation.
>
> 2. Prevalence of instructional documents on the web: we are limited by the information on the web. This has different limitations than what LLMs can generate, such as the inability to find information due to retrieval failures.
>
>
> > The choice of LLM is not really motivated. What is the reason for the choice? How do the findings compare to a different LLM?
>
> We use gpt-4o-mini-2024-07-18 as the LLM for (a) generating queries + URL filtering (b) summarizing instructional documents (c) aggregating across queries and generating criteria. We chose this to balance performance and cost during our experiments, as this was the most performant LLM available at its price point. In our preliminary experiments we also tested gpt-4-turbo-2024-04-09 and claude-3-5-haiku-20241022 along with 4o-mini, but we did not observe any meaningful differences in performance. We can include a discussion of these in any future version of the paper.
>
>
> > The reference to significance testing in Table 2 is not quite clear. The difference is significant compared to what? Also, do you apply a suitable correction such as Bonferroni?
>
> The significance tests in Table 2 take as the null hypothesis that EA-Web is not better than each baseline method along the given metric values. With Bonferroni correction the significance test for all but one (the difference between EA-Web and LLM-n on the Implicitness metric) still holds. We can clarify this in any future version.
>
>
> > Would it not make sense to include another baseline that is not entirely based on LLMs but based on a manual creation of evaluation criteria?
>
> We include a human-written (or human-verified) criteria baseline for each of our evaluations and show the results in Table 2, Table 3 and Figure 4. The criteria are a part of the datasets we considered in this work. For example, in Ask-then-Critique we have users evaluate LLM responses to their own instructions and generate a criteria. Similarly, Dolomites has expert-written human criteria for the instructions.

---

> > ### Comment · Reviewer_tKpV · 2025-06-07
> >
> > I appreciate the effort put in by the authors to address my concerns. Some aspects are now clearer but I still do not see a compelling downstream use case being tested to demonstrate that the findings translate to benefits in a practical real-world application (that is what I would consider an extrinsic evaluation). I also still find that a fair number of assumptions are a bit ad hoc making it difficult to have confidence in the generalisability of the results. I therefore stick to me initial assessment.

---

### Decision · Program_Chairs · 2025-07-08

**Decision:**

Accept

**Comment:**

The author(s) of this paper propose EvalAgent, a framework designed to automatically discover latent task-specific criteria. EvalAgent consists of 3 stages: a query generator, an expert retriever, and a criteria generator. They find that the criteria generated by EvalAgent are more implicit and specific, more actionable, and align better with human-written criteria.

There is disagreement among the reviewers about the paper being ready for publication. The rebuttal phase has been quite intense, and from it, it appears that the most important limitations raised by the reviewers have been answered by the authors in their responses; some of these could be addressed in the main body of the camera-ready, others could be discussed as limitations of the current work.